# HIGH-ACCURACY AND DIMENSION-FREE SAMPLING WITH DIFFUSIONS

## ABSTRACT

Diffusion models have shown remarkable empirical success in sampling from rich multi-modal distributions. Their inference relies on solving a certain differential equation initialized at pure noise. However, this differential equation cannot be solved in closed form, and its resolution via discretization typically requires many small iterations to produce *high-quality* samples. More precisely, prior works have shown that the iteration complexity of discretization methods for diffusion models scales polynomially in the ambient dimension and the inverse accuracy $1/\varepsilon$. In this work, we propose a new solver for diffusion models relying on the collocation method Lee et al. (2018), and we prove that its iteration complexity scales *logarithmically* in $1/\varepsilon$, and does not depend explicitly on the ambient dimension. More precisely, the dimension affects the complexity of our solver through the *effective radius* of the support of the target distribution only. Our solver constitutes the first "high-accuracy" diffusion-based sampler that only uses approximate access to the scores of the data distribution.

## 1 INTRODUCTION

Diffusion models Sohl-Dickstein et al. (2015); Song & Ermon (2019); Ho et al. (2020); Dhariwal & Nichol (2021); Song et al. (2021b;a); Vahdat et al. (2021) are the dominant paradigm in image generation, among other modalities. They sample from high-dimensional distributions by numerically simulating a *reverse process* driven by a certain differential equation with drift learned from training data. The reverse process is meant to undo some noise process, and by simulating the reverse process sufficiently accurately, one can generate fresh samples from the distribution out of pure noise.

The empirical success of this method has spurred a flurry of theoretical work in recent years to understand the mechanisms by which diffusion models are able to easily sample from rich multimodal distributions. These works draw upon tools from the extensive literature on log-concave sampling and arrive at a remarkable conclusion: diffusion models can efficiently sample from *any* distribution in high dimensions, even highly non-log-concave ones, provided one has access to a sufficiently accurate estimate of the score of the distribution along a noise process (see Section 1.2 for an overview of this literature).

The earliest such results showed that for any smooth distribution with bounded second moment, one can sample to error $\varepsilon$ in total variation distance in $\text{poly}(d, 1/\varepsilon)$ iterations, given sufficiently accurate score estimation Lee et al. (2023); Chen et al. (2023c;a). These results focus on samplers that simulate the reverse process in discrete time. The discretization introduces bias, which necessitates taking the discretization steps sufficiently small – i.e. of size $\text{poly}(\varepsilon, 1/d)$ – to ensure the trajectory of the sampler remains sufficiently close to that of the continuous-time reverse process.

While these discretization bounds have been refined significantly by recent work (see Section 1.2), what remains especially poorly understood is the dependence on $\varepsilon$. In the log-concave sampling literature, there is a well-understood taxonomy along this axis: there are (1) "low-accuracy" methods like Langevin Monte Carlo that get iteration complexity scaling polynomially in $1/\varepsilon$, and (2) "high-accuracy" methods which correct for discretization bias via Metropolis adjustment and get iteration complexity *polylogarithmic* in $1/\varepsilon$.

For diffusion models, guarantees of the second flavor are conspicuously absent. While our focus is theoretical, we briefly remark that this is also relevant from a practical perspective, especially

in hardware-constrained settings where one cannot afford to directly modify the neural network generating the score estimates. In these contexts, the most common approach to improving generation quality is simply to take a numerical solver like Euler discretization or more sophisticated variants and drive the step size down. This forces the sampler to take more steps, and the conventional wisdom is that as the number of steps increases, the sampler is able to generate higher-quality results with finer details.

In this work, we make progress towards identifying more efficient ways of sampling with diffusion models in this high-accuracy regime. We therefore ask:

> *Are there samplers for diffusion modeling that achieve $O(\text{polylog} 1/\varepsilon)$ iteration complexity?*

### 1.1 OUR CONTRIBUTIONS

Here we answer this in the affirmative. We will focus on the following class of distributions $q$:

**Assumption 1** (Bounded plus noise). *Let $R, \sigma > 0$. There exists a distribution $q'$ supported on the origin-centered ball of radius $R$ in $\mathbb{R}^d$ for which $q = q' \star \mathcal{N}(0, \sigma^2 I)$.*

In other words, we consider sampling from a distribution $q$ which is the convolution of a compactly supported distribution with Gaussian noise. Our bounds will scale polynomially in $R/\sigma$. A natural example of a distribution satisfying Assumption 1 is a mixture of isotropic Gaussians; more generally, as observed by Chen et al. (2023c), distributions satisfying Assumption 1 naturally model what diffusion models in practice try to sample from via early stopping. Positivity of $\sigma$ also makes it possible to prove convergence guarantees in strong divergence-based metrics like total variation distance.

In this paper, we consider sampling from $q$ by approximately simulating the reverse process. We first show a key structural property: the high-order time derivatives of the score function along the reverse process are bounded, implying that the score function can be pointwise-approximated by a low-degree polynomial in time. The statement is technical and we defer it to the supplement.

Leveraging this result, we show how to adapt a certain specialized ODE solver of Lee et al. (2018) to simulate steps along the reverse process, and argue that our simulation remains close to the trajectory of the reverse process at all times, see Algorithm 2. We thus prove that Algorithm 2 can efficiently sample from $q$, yielding the following main guarantee:

**Theorem 2** (Informal, see Corollary 14). *Let $\varepsilon, R, \sigma > 0$. Suppose $q$ is a distribution satisfying Assumption 1 for parameters $R, \sigma$, and suppose the true score is $L$-Lipschitz (see Assumption 28). Given access to score estimates along the forward process that are Lipschitz in the space variable, for which the estimation error is $L^2$-bounded and has subexponential tails, there is a diffusion-based sampler (see Algorithm 2) which outputs samples from a distribution $\varepsilon$ close to $q$ in total variation (TV) distance in $(R/\sigma)^2 \cdot \log 1/\varepsilon$ iterations.*

To the best of our knowledge, this is the first diffusion-based sampler to achieve logarithmic, rather than polynomial, dependence on $1/\varepsilon$, while only assuming access to a sufficiently accurate score. While our result requires a stronger assumption on the distribution of score error than prior work, we note that the analysis in prior work incurs polynomial dependence on $1/\varepsilon$ even assuming *perfect* score estimation.

Another appealing feature of our result is that the iteration complexity of our algorithm is *independent* of the ambient dimension $d$, instead depending on the radius $R$ in Assumption 1. While in general $R$ can scale with $d$, the aforementioned Gaussian mixture setting offers a natural setting where $R$ can be much smaller than $d$. For example, in the theory of distribution learning, the most challenging regime is precisely when the component centers are at distance $\Theta(\sqrt{\log k})$ from each other (Hopkins & Li, 2018; Kothari et al., 2018; Diakonikolas et al., 2018; Liu & Li, 2022). This corresponds to the case of $R = \Theta(\sqrt{\log k})$ and $\sigma = 1$, for which our sampler achieves iteration complexity scaling only polylogarithmically in $k$ and $1/\varepsilon$, whereas existing diffusion-based methods would scale polynomially in $d$ and $1/\varepsilon$ for this example (see Section 1.2 for discussion of a concurrent result (Li et al., 2025) that also obtains dimension-free results for sampling from Gaussian mixtures).

## 1.2 Related work

**Theory for diffusion models.** We review the extensive line of recent works on diffusion model convergence bounds in Appendix A, focusing here on the threads most relevant to our result.

First, to our knowledge, the only prior work which studies the question of high-accuracy sampling for diffusions is Huang et al. (2024a). While they also achieve exponentially improved $\varepsilon$ dependence using MALA-style methods, they work in a stronger access model than is commonly studied in this literature, as they assume approximate query access to log density ratios between arbitrary points, in addition to the scores.

As for our dependence on $R/\sigma$ instead of $d$, there is a relevant line of work Li & Yan (2024a); Liang et al. (2025); Huang et al. (2024b); Potaptchik et al. (2024) on sampling from distributions with low intrinsic dimension. They show that for any distribution whose intrinsic dimension, which they quantify in terms of covering number of the support, is $k$, DDPM can sample with iteration complexity $O(k^4/\varepsilon^2)$. While this provides another example of a diffusion-based sampler whose rate adapts to the underlying geometry of the distribution, it is incomparable to our result: there can be supports which are high-dimensional but bounded in radius, and vice versa.

Finally, we note that the collocation method (see Section 2.3) has been studied in the context of diffusions, but primarily as a way to *parallelize* the steps of the sampler Anari et al. (2023); Gupta et al. (2024); Chen et al. (2024a), but not using low-degree polynomial approximation. The key difference relative to our work is that by showing low degree approximability of the score in time, we can prove that the implementation of collocation by Lee et al. (2018) via low-degree polynomials over a small time window of the probability flow ODE converges at an *exponential* rate.

**High-accuracy sampling.** The literature on high-accuracy samplers, which is traditionally centered around log-concave distributions and more generally distributions which satisfy a functional inequality, is too extensive to do justice to here, and we refer to Chapter 7 of Chewi (2023) for a detailed overview. We briefly overview this in Appendix A.

Our techniques do not draw upon this literature but are instead based on the work of Lee et al. (2018), which devised the general framework of collocation via low-degree approximation that we employ. Motivated by sampling problems connected to logistic regression, they applied their method to obtain a high-accuracy sampler for densities of the form $q \propto e^{-f}$, where $f(x) = \sum_i \phi_i(\langle a_i, x \rangle) + \lambda \|x\|^2$.

**Independent work.** Recently, Li et al. (2025) considered the special case of mixtures of isotropic Gaussians and showed using very different techniques that diffusion models can sample from them in $T = \tilde{O}(\text{polylog}(k)/\varepsilon)$ iterations, provided that the means of the Gaussians have norm at most $T^C$ for arbitrarily large absolute constant $C$. This result is incomparable to ours. We obtain polylogarithmic dependence on $1/\varepsilon$ and our guarantee applies to a wide class of distributions beyond just the special case of mixtures of isotropic Gaussians. On the other hand, the result of Li et al. (2025) improves upon our guarantee for Gaussian mixtures: whereas we have a polynomial dependence on the maximum norm of any center in the mixture, they have an *arbitrarily small* dependence on the radius.

## 1.3 Roadmap

In Section 2, we provide technical preliminaries and give a description of our sampler. In this section we also provide intuition for how this sampler can achieve the high-accuracy guarantee of Theorem 2. In Section 3, we sketch the proof of our main structural result showing that the drift of the reverse process is well-approximated by a low-degree polynomial in time. In Section 4, we sketch the main steps for completing the proof of Theorem 2. We defer the full proof details to the supplement.

## 2 Preliminaries and our algorithm

**Notation.** Let $\gamma^d$ denote the $d$-dimensional standard Gaussian distribution. For any distribution $p$ on $\mathbb{R}^d$, $L^2(p)$ denotes the space of squared integrable random vectors in $\mathbb{R}^d$. We recall the definitions of the Total Variation distance $\text{TV}(P, Q) = \frac{1}{2} \int_{\mathbb{R}^d} |p - q| \, dx$ for densities $p, q$, and the Wasserstein-2 distance $W_2^2(P, Q) = \inf_{\pi \in \Pi(P,Q)} \int \|x - y\|^2 \, d\pi(x, y)$ where $\Pi(P, Q)$ is the set of

couplings. Finally, for any vector $x = (x_1, \ldots, x_d) \in \mathbb{R}^d$, $\|x\|_\infty = \max_i |x_i|$ and if $x$ is random, $\|x\|_{p,\infty} = \max_i \left(\mathbb{E}|x_i|^p\right)^{1/p}$.

## 2.1 DIFFUSION MODELS

Here we review the basics of diffusion models; for a more detailed overview, we refer the reader to the surveys of Chen et al. (2024b) and Nakkiran et al. (2024).

Diffusion models are built upon two main components: the *forward process* and the *reverse process*. The forward process is a noise process driven by a stochastic differential equation of the form

$$\mathrm{d}x_t = \frac{\dot{\alpha}_t}{\alpha_t} x_t \, \mathrm{d}t + \alpha_t \sqrt{2\beta_t \dot{\beta}_t} \, \mathrm{d}B_t, \qquad x_0 \sim q',$$

where $\alpha_t, \beta_t \geq 0$ are time-dependent parameters that one can choose, and $(B_t)_{t\geq 0}$ is a standard Brownian motion in $\mathbb{R}^d$. Conditioned on $x_0$, the process at time $t$ is distributed as $\alpha_t(x_0 + \beta_t g)$ for $g \sim \gamma^d$; denote this marginal distribution by $p_t$, where $p_0 = q'$. It is common to choose $\alpha_t = e^{-t}$ and $\beta_t = \sqrt{e^{2t} - 1}$ so that the forward process is the standard OU process with stationary distribution $\gamma^d$. We run the forward process up to time $T$. Define $\sigma_t = \alpha_{T-t}\beta_{T-t}$.

The *reverse process* is designed to undo this noise process, i.e. transform $p_T$ to $p_0$. For convenience of the notation, we denote $p_{T-t}$ by $q_t$. One version of this reverse process is given by the *probability flow ODE*, which is specified by

$$\mathrm{d}y_t = \left[ \frac{\dot{\alpha}_t}{\alpha_t} y_t + \alpha_t^2 \beta_t \dot{\beta}_t \nabla \ln q_t(y_t) \right] \mathrm{d}t, \tag{1}$$

where $\nabla \ln q_t$ is called the *score function*. The key property is that if $x_0 \sim q_0$, then $x_T \sim q_T = q'$. In practice, this is run using *score estimates* $s_t \approx \nabla \ln q_t$ instead of the actual score functions, and in the theoretical literature it is standard to assume that these are close in $L_2(q_t)$. In this work, we will make a somewhat stronger assumption:[1]

**Assumption 3.** *We assume the error incurred by the* score estimate $s_t : \mathbb{R}^d \to \mathbb{R}^d$ *has sub-exponential norm $\varepsilon_{err}$ for all $0 \leq t \leq T$. That is, $\mathbb{P}_t[\|s_t - \nabla \ln q_t\| \geq z] \leq 2\exp(-z/\varepsilon_{err})$ for all $y \geq 0$.*

Additionally, instead of initializing the ODE at $y_0 \sim q_0$, one initializes at $y_0 \sim \gamma^d$ for some noise distribution for which $\pi \approx q_0$ and from which it is easy to sample. For instance, in the example of the standard OU process, we can take $\pi = \gamma^d$ because the forward process converges exponentially quickly to $\gamma^d$. As $\pi$ is easy to sample from, the probability flow ODE can be used to (approximately) generate fresh samples from $q$.

We further assume the score estimate is $\tilde{L}$-Lipschitz.

**Assumption 4.** $s_t$ *is $\tilde{L}$-Lipschitz:* $\forall x, y \in \mathbb{R}^d$, $\|s_t(x) - s_t(y)\| \leq \tilde{L}\|x - y\|$.

## 2.2 A FUNDAMENTALLY DIFFERENT VIEW ON DISCRETIZATION

Traditionally, to simulate the continuous-time ODE in discrete time, some numerical method like Euler-Maruyama discretization is used. Such discretization schemes can only approximately simulate the continuous time ODE by using a fixed gradient in a window of time. For example, to approximate the ODE $x_t = f_t(x_t)dt$ with one gradient step in time window $[0, h]$, the Euler-Maruyama method uses the gradient at the beginning time 0, resulting in the update $\hat{x}_h = hf_0(\hat{x}_0) + x_0$, where $\hat{x}_h$ denotes the discretized process. Here, $h$ is the step size of the algorithm. Unfortunately, this type of discretization puts a fundamental dimension-dependent limitation on how large the step size can be.

To illustrate the point, consider the vector field $f_t = x$ for all times $t \geq 0$. In this case, the discretized step is given by $\hat{x}_h = h(1)_{i=1}^d + x_0 = ((1 + h))_{i=1}^d$ while the solution of the ODE $x_t = f_t(x_t)dt$ with initial condition $x_0 = (1)_{i=1}^d$ is given by $x_t = (e^t)_{i=1}^d$. This implies $\|x_h - \hat{x}_h\| =$

---

[1] Intuitively, the reason we need to make a stronger assumption on the tails of the score estimation error is as follows. If the tails were heavier, then under the event in which the score estimate differs appreciably from the true score, the sampler can deviate sufficiently that it will fail to converge in $\mathrm{polylog}(1/\varepsilon)$ many steps. We suspect that this is fundamental and leave proving a suitable lower bound as an open question.

$\|(e^h - (1 + h))_{i=1}^d\| = \Theta(h^2 \sqrt{d})$. Therefore, in order to keep the deviation $\|\hat{x}_h - x_h\|$ bounded by $O(1)$, the step size $h$ has to be $O(1/d^{1/4})$. This is because as we move along the ODE solution $x(t)$, the vector field $f_t(x_t)$ can change drastically from its initial value, so using a fixed vector $f_0(x_0)$ as the velocity does not allow us to go that far.

However, if we had a prior knowledge about the functional form of $f_t(x_t)$ along the path $(x_t)_{[0,h]}$, perhaps we could use that to take larger steps to simulate the ODE. For example, if we knew $f_t(x_t)$ lives in a low-dimensional subspace with basis $\{\phi_j(t)\}_{j=1}^D$, namely $f_t(x_t) = \sum_{i=1}^D c_i \phi_i$ for some coefficients $(c_i)_{i=1}^D \in \mathbb{R}^D$, then if we could estimate these coefficients, that allows us to predict the path $x_t$ for large times $t$.

In this work, we show that perhaps suprisingly, under minimal conditions on the target distribution $q'$ each coordinate of the score function on the probability flow ODE path, $f_t(x_t)$, is well-approximated by a low-degree polynomial in $t$. We then prove that one can estimate these coefficients for each coordinate using the *collocation method* together with our score estimates. Notably, this enables us to take longer discretized steps, without any explicit dimension dependency, to follow the ODE path. We emphasize that while the collocation method, i.e. Picard iteration, has been used in prior diffusion model theory work Gupta et al. (2024); Chen et al. (2024a); Shih et al. (2024), the key novelty in our approach is to leverage low-degree polynomial approximation.

## 2.3 COLLOCATION METHOD

The *collocation method* is a numerical scheme for approximating the solution to an ordinary differential equation through fixed-point iteration. Here, consider a generic initial value problem: $\mathrm{d}x_t = f_t(x_t)\,\mathrm{d}t$ and $x_0 = v$ for all $t \in [0, H]$. This admits the integral representation $x_t = v + \int_0^t f_s(x_s)\,\mathrm{d}s$, which can be thought of as the fixed-point solution to the equation $x = \mathcal{T}(x)$ for the operator $\mathcal{T} : \mathcal{C}([0,H], \mathbb{R}^d) \to \mathcal{C}([0,H], \mathbb{R}^d)$ given by $\mathcal{T}(x)_t \triangleq v + \int_0^t f_s(x_s)\,\mathrm{d}s$ which has a unique fixed point provided there exists $k \in \mathbb{N}$ such that $\mathcal{T}^k$ is $L$-Lipschitz with $L < 1$. Authors in Lee et al. (2018) proposed to solve this fixed-point equation as follows. Suppose that each coordinate of the time derivative of the solution, i.e. $t \mapsto f_t(x_t)$, is well-approximated by a low-degree polynomial. Letting $\phi_1, \dots, \phi_D : [0, H] \to \mathbb{R}$ be an appropriately chosen basis of polynomials, by polynomial interpolation we can find nodes $c_1, \dots, c_D$ such that $\phi_j(c_i) = \mathbb{1}[i = j]$ for all $i, j \in [D]$ so that in particular,

$$\frac{\mathrm{d}x_t}{\mathrm{d}t} \approx \sum_{j=1}^D f_{c_j}(x_{c_j})\phi_j(t)\,.$$

Writing this in integral form as before, we arrive at the approximate fixed-point equation:

$$x \approx \mathcal{T}_\phi(x)\,, \qquad \mathcal{T}_\phi(x)_t \triangleq v + \int_0^t \sum_{j=1}^D f_{c_j}(x_{c_j})\phi_j(s)\,\mathrm{d}s\,. \tag{2}$$

This suggests a natural algorithm: instead of maintaining the entire continuous-time solution $x : [0, H] \to \mathbb{R}^d$ over the course of fixed-point iteration, simply maintain the values $(x_{c_j})_{j \in [D]}$ and update these according to Eq. (2), which amounts to a matrix-vector multiplication. This algorithm is summarized in Algorithm 1 below.

Lee et al. (2018) give conditions under which collocation with appropriately chosen basis polynomials and nodes converges to a sufficiently accurate solution to the ODE. We will need to adapt their guarantees to our setting to account for score estimation error. The full guarantee will be given in Section 3. We require the following condition for the polynomial basis $\phi$:

**Definition 5.** *We say that $\phi$ is $\gamma$-bounded if, for all basis elements $\phi_j : [0, H] \to \mathbb{R}$ and $t \leq H$,* $\sum_{j=1}^D \left| \int_0^t \phi_j(s)\,\mathrm{d}s \right| \leq \gamma t$.

## 2.4 OUR ALGORITHM

We are now ready to combine the ingredients in the previous subsections to give a description of our sampler. The algorithm simply splits up the reverse process into a series of small time windows, each

---

**Algorithm 1:** PICARD($(f_t)_{t\in[0,H]}, v, (c_j)_{j=1}^D N$)

---

**Input:** Vector field $f_t : \mathbb{R}^d \to \mathbb{R}^d$ for $t \in [0, H]$, initial value $v$, number of iterations $N$
**Output:** Approximate solution $\hat{x}_H$ to initial value problem
1 Define $A_\phi \in \mathbb{R}^{D \times D}$ by $(A_\phi)_{i,j} = \int_0^{c_j} \phi_i(s) \, ds$
2 Define $X^{(0)} \in \mathbb{R}^{d \times D}$ to be $v1_D^\top$, where $1_D$ is the all-ones vector.
3 **for** $t = 0, \ldots, N-1$ **do**
4 $\quad$ Define $F_c(X^{(t)}) \in \mathbb{R}^{d \times D}$ by $F_c(X^{(t)})_{:,j} = f_{c_j}(X_{:,j}^{(t)})$
5 $\quad$ $X^{(t+1)} \leftarrow v1_D^\top + F_c(X^{(t)})A_\phi$ $\qquad\qquad\qquad\qquad$ // (2)
6 **end**
7 **return** $v + \int_0^H \sum_{i=1}^D F_{c_i}(X_{:,i}^{(N)})\phi_i(s) \, ds$

---

---

**Algorithm 2:** COLLOCATIONDIFFUSION

---

**Input:** Score estimates $s_t$ satisfying Assumption 3, target error $0 < \varepsilon < 1$, denoising schedule
$\qquad 0 < t_1 < \cdots < t_n = T$, Picard depth $N$
**Output:** Sample from a distribution $\hat{q}$ satisfying $W_2(\hat{q}, q) \le \varepsilon$
1 $\hat{x}_0 \sim \mathcal{N}(0, \sigma^2)$
2 **for** $k = q, \ldots, n-1, n$ **do**
3 $\quad$ $\hat{x}_{t_k} = $ PICARD($(s_t)_{t\in[t_{k-1}, t_k]}, \hat{x}_{t_{k-1}}, N$)
4 **end**
5 **return** $\hat{x}_{t_n}$

---

of which is of length $\tilde{O}(\frac{\sigma^2}{R^2})$, and successfully solves the corresponding initial value problems in each of these windows using collocation via PICARD. There will be some accumulation of errors, as the PICARD calls do not result in exact solutions, but these errors can be made extremely small because PICARD solves the relevant ODEs to extremely high precision.

## 3 LOW-DEGREE APPROXIMATION OF THE SCORE

In this section we describe the main ideas behind our proof. Recall that key to our approach is to prove a structural result showing that the time derivative of the score function along the reverse process can be controlled.

### 3.1 LOW-DEGREE APPROXIMATION ALONG THE TRUE REVERSE PROCESS

Our starting point is to relate time derivatives of the score at a point $y_t$ along the reverse process to higher-order moments of the posterior distribution on the sample $y_T$ that would have generated $y_t$ along the forward process.

We begin with the following calculation.

**Lemma 6.** *Suppose $y_t = X_t + \sigma_t \xi$, where $X_t = \alpha_t \bar{X}$ for $\bar{X} \sim q'$ and $\xi \sim N(0, I)$. We use the notation $\mathbb{E}^{t,y_t}$ to denote the conditional expectation of $X_t$ given $y = X_t + \sigma_t \xi$. Define the posterior mean $\mu_t(y) \triangleq \mathbb{E}^{t,y} X_t$.*

*The time derivative of the vector field of probability flow ODE can be calculated as*

$$\partial_t(y_t + \nabla \log q_t(y_t)) = \left( y_t + \frac{\mathbb{E}^{t,y_t} X_t - y_t}{\sigma_t^2} \right)$$

$$+ \frac{1}{\sigma_t^2}\mathbb{E}^{t,y_t} X_t + \frac{1-\sigma_t^2}{2\sigma_t^6}\mathbb{E}^{t,y_t}\left( \langle y_t, X_t \rangle + \|X_t\|^2 \right)(X_t - X_t') + \frac{4}{\sigma_t^4}\mathbb{E}^{t,y_t}\left( \langle y_t, X_t \rangle - \|X\|^2 \right)(X_t - X_t')$$

$$+ \frac{1}{\sigma_t^2}\left(\frac{1}{\sigma_t^2} - 1\right)y_t - \frac{1}{\sigma_t^4}\mathbb{E}^{t,y_t} X_t + \frac{1}{\sigma_t^4}\mathbb{E}^{t,y_t}\left( 2\left\langle y_t, \frac{X_t''}{\sigma_t^2} \right\rangle - \langle X_t, y_t \rangle - \frac{1}{\sigma_t^2}\langle X_t, X_t'' \rangle \right)(X_t - X_t').$$

Iterating this calculation multiple times, we arrive upon the following key calculation expressing the higher-order derivatives of the vector field of the probability flow ODE:

**Lemma 7.** *The $k$th derivative of the backward ODE vector field can be expanded as*

$$\partial_t^k \left( y_t + \nabla \ln q_t(y_t) \right) \tag{3}$$

$$= \sum_{r_1, r_2, r_3, r_4 \leq k, \, \mathbf{i} \in [k]^{r_3}, \mathbf{k} \in [k]^{r_4}, \mathbf{l} \in [k]^{r_4}} \frac{\left(1 - \sigma_t^2\right)^{r_1}}{\sigma_t^{r_2}} \mathbb{E}^{t,y_t} \prod_{j=1}^{r_3} \left\langle y_t, X^{(\mathbf{i}_j)} \right\rangle \prod_{j=1}^{r_4} \left\langle X^{(\mathbf{k}_j)}, X^{(\mathbf{l}_j)} \right\rangle (a_{\mathbf{i}, \mathbf{k}, \mathbf{l}} X + b_{\mathbf{i}, \mathbf{k}, \mathbf{l}} y_t) \tag{4}$$

*where $r_1 \leq k, r_2 \leq 4k$ and $\sum_{\mathbf{i} \in [k]^{r_3}, \mathbf{k} \in [k]^{r_4}, \mathbf{l} \in [k]^{r_4}} \left| a_{\mathbf{i}, \mathbf{k}, \mathbf{l}} \right| + \left| b_{\mathbf{i}, \mathbf{k}, \mathbf{l}} \right| \leq (74k)^k$. Here the random variables $X^{(i)}$ are independently distributed according to the posterior distribution $q^{t,y}$, and $\mathbf{i}, \mathbf{k}, \mathbf{l}$ are arbitrary tuples of indices.*

This result can be understood in the same spirit as Tweedie's formula, which relates the posterior mean of $X$ to the score function. Higher-order generalizations of Tweedie's formula in the literature Meng et al. (2021) typically relate moments of the posterior to derivatives of the score *with respect to the space variable*, whereas in Lemma 7 we consider derivatives with respect to the *time variable*.

An important feature of our technique is to bound this multiple integration formula for higher derivatives of the score independent of the ambient dimension $d$, which will be crucial for obtaining iteration complexity bounds that only depend on the effective radius $R$ of the distribution. More precisely, the bound on the effective radius allows us to bound expectations of the form $\mathbb{E}[\prod_{j=1}^{r_4} \langle X^{(\mathbf{k}_j)}, X^{(\mathbf{l}_j)} \rangle]$ pointwise and expectations of the form $\mathbb{E}[\prod_{j=1}^{r_3} \langle y, X^{(\mathbf{i}_i)} \rangle]$ with high probability by dimension-independent quantities, from which we can deduce bounds on the left-hand side of Lemma 7 via applications of Cauchy-Schwarz (see Lemma 19). Leveraging these formulas, we arrive at our first key technical ingredient, a bound on the higher-order time derivatives of the score:

**Lemma 8.** *If $q$ satisfies Assumption 1, $(y_t)_t$ is a solution to (1), and $y_0 \sim q_T$, then the $k^{th}$ derivative of the vector field along the probability flow ODE is bounded as*

$$\left\| \partial_t^k \left( y_t + \nabla \log q_t(y_t) \right) \right\|_{p,\infty} \lesssim R \left( \frac{74k}{\sigma_t^2} \right)^k \left( \frac{R}{\sigma_t} + (kp)^{1/2} \right)^{2k}$$

Note that these bounds are singly exponential in the order of the derivative and scale with $(R/\sigma_t)^k$. If these derivative bounds held pointwise, then we could simply appeal to the Taylor remainder theorem to argue that the true score is well-approximated by a low-degree polynomial. But ultimately Lemma 8 only offers a high-probability statement over the distribution induced by the true reverse process. We will need to handle deviations from the true reverse process, which arise because the sampler is initialized at Gaussian instead of the correct marginal, and because in each subsequent window in which we apply collocation, the initialization is also slightly off from the correct marginal.

### 3.2 LOW-DEGREE APPROXIMATION ALONG THE ALGORITHM

To argue about low-degree approximation along the trajectory of the actual sampler, we will need to argue that the posterior distribution from Lemma 7 is robust to perturbations to the conditioning.

Our key tool for doing so is the following coupling lemma:

**Lemma 9** (Coupling). *Let $\tilde{y}$ a random variable such that $\|y - \tilde{y}\| \leq \delta$ for $\delta \leq \frac{1}{6(\sqrt{d} + \sqrt{\ln(1/\varepsilon_1)})}$. Now consider the conditional distribution $p(.|\tilde{y})$, where $p(x, y)$ is the joint distribution of $(X_t, y_t)$, for $y_t = X_t + \xi$, $X_t = \sqrt{1 - \sigma_t^2} \bar{X}$, $\bar{X} \sim q'$, and $\xi \sim N(0, \sigma_t^2 I)$. Then, given $T - t \geq 1$, under the event $\mathcal{A}_1$ defined in Equation (16), we have*

$$TV(p(.|y), p(.|\tilde{y})) \leq 8\delta(\sqrt{d} + \sqrt{\ln(1/\varepsilon_1)}) \tag{5}$$

*In particular, Inequality (5) holds with probability at least $1 - \varepsilon_1$.*

Combining Lemmas 9 and 8, we obtain a bound on higher time derivatives of the score along the reverse process when initialized at a distribution that is slightly off from the true marginal:

**Theorem 10.** *Let $\delta \in (0, 1)$. If $q$ satisfies Assumption 1, and $(\tilde{y}_t)_t$ is given by initializing the reverse process at a distribution which is $\delta$ $W_2$-**close** to $q_0$ and running the probability flow ODE. Then, for $\delta$ small enough (as rigorously characterized in Lemma 25), with probability at least $1 - \delta$ over the randomness of the initialization, the $k^{th}$ derivative of the vector field along the probability flow ODE is bounded at $\tilde{y}_t$ by*

$$\left\|\partial_t^k \left(\tilde{y}_t + \nabla \log q_t(\tilde{y}_t)\right)\right\|_\infty \lesssim R \left(\frac{k}{\sigma_t^2}\right)^k \left(\frac{R}{\sigma_t}\right)^{2k} + (k \log(74kd/\delta))^{2k} \ .$$

This theorem provides the theoretical underpinning for our approach. Because it is a high probability bound over iterates of the actual sampler, we can now instantiate the aforementioned Taylor truncation argument to get the desired low-degree approximation along the trajectory of the sampler itself, rather than of the idealized reverse process. We will use this ingredient in conjunction with the technology in Lee et al. (2018) to prove the following convergence bound for PICARD (Algorithm 1):

**Proposition 11** (Informal, see Appendix). *Let our score estimate be $\tilde{L}$ Lipschitz. Given that $\tilde{L}\gamma_\phi h \leq \frac{1}{2}$, then for the correct backward probabiliy flow ODE $y_t$ and arbitrary $x \in \mathcal{C}([t_0, t_0 + h], \mathbb{R}^d)$,*

$$\left\|T_\phi^{\circ m}(y) - y\right\|_{[t_0, t_0+h]} \leq 2(\varepsilon_{ld} + \max_{1 \leq j \leq D} \left\|s_{c_j}(y(c_j)) - y(c_j) - \nabla \log q_{c_j}(y(c_j))\right\|)(1 + \gamma_\phi)h$$

$$\left\|T_\phi^{\circ m}(x) - T_\phi^{\circ m}(y)\right\|_{[t_0, t_0+h]} \leq \frac{1}{2^m} \left\|x - y\right\|_{[t_0, t_0+h]} \ ,$$

*where here $\|\cdot\|_{[a,b]}$ denotes the sup norm over the interval $[a, b]$, and $\varepsilon_{ld}$ denotes the approximation error (in sup norm) incurred by approximating the score function along the probability flow ODE with a polynomial in the span of $\phi$. See section 2.3 for the definition of $c_j$'s.*

Note that the solver can only be run for time which scales inversely in the boundedness $\gamma$ of the basis (recall Definition 5), which can be of constant order, and inversely in the Lipschitzness of the vector field of the probability flow ODE. This is why we cannot directly use collocation to solve the probability flow ODE in one shot. Nevertheless, the crucial point is that, as the second inequality above suggests, the error incurred by PICARD is decreasing exponentially in the number of iterations, modulo some additional error that needs to be carried around coming from low-degree approximation and score estimation error.

We note that while the core idea for the Proposition is from Lee et al. (2018), one novelty of our bound is that we account for score estimation error and show that the errors coming from that do not compound exponentially over the Picard iterations. This will be essential in the sequel for keeping our sampler sufficiently close to the true trajectory of the probability flow ODE.

## 4 CONVERGENCE OF THE SAMPLER

We now have all the necessary ingredients to prove our main result.

We first show how to get a sampler which is $W_2$-close to $q$. An appealing feature of this guarantee relative to our TV convergence guarantee is that 1) the requirement on the score estimation error is far less stringent, and 2) it only involves running the algorithm COLLOCATIONDIFFUSION, i.e. no postprocessing is needed on top of simply simulating the true reverse process using the ODE solver of Lee et al. (2018) to implement discrete steps.

**Theorem 12.** *Let $\varepsilon_{err} \geq \varepsilon$, $\sigma > 0$ and $R \geq 1$. Suppose $q$ is a distribution satisfying Assumption 1 for parameters $R, \sigma$. Given access to score estimates $s_t$ satisfying Assumption 3, with probability at least $1 - O(\varepsilon(R/\sigma)^2)$ Algorithm 2 outputs samples from a distribution $\hat{q}$ for which $W_2(q, \hat{q}) \leq \tilde{O}(\varepsilon)$ in $\tilde{O}((R/\sigma)^2)$ number of rounds and $\tilde{O}(1)$ number of Picard iterations per round.*

**Remark 13.** *$\tilde{O}$ hides polylogarithmic factors in $1/\varepsilon, d, R$. Furthermore, the assumption $R \geq 1$ is only for simplifying the bounds. For the full theorem please see Appendix G.*

Here we informally sketch the proof of this result, deferring the proof details to the Appendix. The idea will be to try tracking the trajectory of the true probability flow ODE closely at all times and successively running PICARD over small time windows.

Suppose that up to some time $t$ in the simulation of the reverse process, the iterate of the sampler is distributed according to a distribution whose $W_2$ distance from the true marginal is small. $q_t$ Then consider trying to simulate the reverse process for another $h$ time steps to approximate $q_{T-t-h}$. One can do this by simply running COLLOCATIONDIFFUSION, provided $h$ is small enough, and because the starting distribution is $W_2$-close to the true marginal, the Taylor truncation argument in conjunction with the bound on the higher-order derivatives of the vector field ensure that we can safely invoke Proposition 11 and driving any error coming from the $W_2$ discrepancy exponentially quickly to zero, leaving behind a fixed amount of score estimation and polynomial approximation error.

Note that the exponential contraction of the error from the $W_2$ discrepancy is absolutely essential to our result. It allows us to "reset" the amount by which we stray from the curve of the true reverse process every time we run PICARD. Interestingly, this is reminiscent of a different recent approach to analyzing the probability flow ODE by Chen et al. (2023b). In that work, the authors tried simulating the ODE for short windows of time, appealing to a naive Wasserstein coupling argument in each of those windows. In their setting, the Wasserstein couplings incur some error that they would like to linearly, not exponentially, accumulate over the course of the sampler. The way to do this was to effectively "restart" the Wasserstein coupling at the start of each new time window by injecting a small amount of noise into the sampler.

**Upgrading to total variation closeness.** Finally, as a consequence of regularizing properties of underdamped Langevin dynamics, our sampling guarantee in Wasserstein distance (Theorem 12) can be converted into a guarantee in total variation distance using standard arguments Gupta et al. (2024); Chen et al. (2024c):

**Corollary 14.** *Let $\varepsilon, R, \sigma > 0$. Suppose $q$ is a distribution satisfying Assumption 1 for parameters $R, \sigma$ and that $\nabla \ln q$ is $L$-Lipschitz. Given access to score estimates $s_t$ satisfying Assumption 3, with $\varepsilon_{sc} \leq \tilde{O}(\frac{\varepsilon^2 L^{1/4}(R/\sigma)^{1/2}}{d^{1/2}})$, Algorithm 3 outputs samples from a distribution $\hat{q}$ for which $\mathrm{TV}(\hat{q}, q) \leq \varepsilon$ in $\tilde{O}\left((R/\sigma)^2\right)$ iterations.*

In the high-accuracy regime, it might seem counterintuitive how one can so easily convert to closeness in TV starting from closeness in Wasserstein, given that one has to run underdamped Langevin Monte Carlo without any kind of Metropolis adjustment. The key idea is to simply *run underdamped Langevin for a shorter amount of time* so that the bias coming from discretization is not too large. We make this argument formal in Appendix H.2.

## 5 CONCLUSION

In this work, we gave a diffusion-based sampler for sampling from a wide class of distributions that includes Gaussian mixtures as a special case, which both has iteration complexity which scales logarithmically in $1/\varepsilon$ ("high-accuracy") and does not depend explicitly on the dimension but rather on the effective radius of the distribution. The key technical step was to establish a bound on the higher-order derivatives of the vector field of the probability flow ODE, which in turn allowed us to approximate it by a low-degree polynomial. This approximation result made it possible to exploit a variant of the collocation method, a fixed-point iteration for numerically solving ODEs, proposed by Lee et al. (2018) which converges exponentially quickly to the true ODE solution if run over sufficiently small time windows.

Our work has a few limitations that raise interesting questions for future study. Firstly, can we somehow relax our assumption on the underlying distribution $q$? Instead of insisting that it is a noised version of a distribution $q'$ whose support is compact, we could merely ask that $q'$ have bounded moments. We could also ask for TV or KL convergence to $q$ in a number of steps which scales only logarithmically in $1/\sigma$, matching what is known in the low-accuracy regime Chen et al. (2023a); Benton et al. (2023). Additionally, it would be interesting to understand whether other strategies for high-accuracy sampling could be used in place of the collocation method that we studied here. For instance, what if one simply ran the probability flow ODE with Euler discretization, but inserted Metropolis adjustment steps after each iteration? Lastly, is the assumption that the score errors have sub-exponential tails truly necessary? We strongly suspect that this is the case and conjecture a query complexity lower bound of $\mathrm{poly}(1/\varepsilon)$ should hold if one only has $L^2$-accurate but heavy-tailed score estimation.

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
