TECHNICAL APPENDICES AND SUPPLEMENTARY MATERIAL

CONTENTS

## A    FURTHER RELATED WORK

**Additional works on discretization bounds for diffusions.**    There have been a large number works in recent years giving general convergence guarantees for diffusion models De Bortoli et al. (2021); Block et al. (2022); De Bortoli (2022); Lee et al. (2022); Liu et al. (2022); Pidstrigach (2022); Wibisono & Yang (2022); Chen et al. (2023c;d); Lee et al. (2023); Li et al. (2023); Benton et al. (2023); Chen et al. (2023b); Benton et al. (2024); Chen et al. (2023a); Gupta et al. (2023); Li et al. (2024b); Li & Yan (2024b); Conforti et al. (2023); Gupta et al. (2024). Of these, one line of work Chen et al. (2023c); Lee et al. (2023); Chen et al. (2023a); Benton et al. (2024); Conforti et al. (2023); Li & Yan (2024b) analyzed DDPM, the stochastic analogue of the probability flow ODE, and showed $\tilde{O}(d)$ iteration complexity bounds, which were recently established by Benton et al. (2024); Conforti et al. (2023) under only the assumption that the data distribution has bounded second moment and very recently refined by Li & Yan (2024b) to apply to distributions with bounded first moment and to depend only linearly on $1/\varepsilon$. Another set of works Chen et al. (2023b;d); Li et al. (2023; 2024a); Gupta et al. (2024); Li et al. (2024b); Li & Jiao (2024); Jiao & Li (2024) studied the probability flow ODE and proved similar rates, one notable difference being that under smoothness assumptions it is known how to get sublinear in $d$ dependence Chen et al. (2023b); Gupta et al. (2024); Li & Jiao (2024); Jiao & Li (2024). As it stands however, all of these works work only in the "low-accuracy" regime, that is, the best-known $\varepsilon$ achieved by these works is still $\mathrm{poly}(1/\varepsilon)$, even under smoothness assumptions.

**High-accuracy sampling for log-concave distributions.**    One of the central ideas in the line of work on high-accuracy log-concave sampling is to append a Metropolis-Hastings filter step after every step of an otherwise low-accuracy sampler. Combining this with Langevin Monte Carlo or Hamiltonian Monte Carlo gives rise to the popular methods of MALA or MHMC respectively. A line of works Dwivedi et al. (2019); Chen et al. (2020); Lee et al. (2020); Chewi et al. (2021); Wu et al. (2022), culminating in the recent breakthrough of Altschuler & Chewi (2024), has established an iteration complexity of $\tilde{O}(\sqrt{d}\mathrm{poly}(1/\varepsilon))$ for MALA for sampling log-concave distributions.

## B    NOTATION

We will work with a data distribution $q$ over $\mathbb{R}^d$.    Throughout, $\bar{X}$ will denote a random vector sampled from $q$, and $X_t$ will denote the scaling $X_t = \sqrt{1 - \sigma_t^2}\bar{X}$. Likewise, we use $\bar{x}$ and $x_t$ to denote realizations of these random vectors. We use $\xi$ to denote a random vector sampled from $N(0, I)$. Given $t, y$, we will use $q^{t,y}$ to denote the posterior distribution on $X_t$ conditioned on $\sqrt{1 - \sigma_t^2}\bar{X} + \sigma_t\xi = y$. We will use $\mathbb{E}^{t,y}$ to denote expectation with respect to $q^{t,y}$.

We will use the following noise schedule

$$\sigma_t \triangleq \sqrt{1 - e^{-2(T-t)}}, . \tag{6}$$

Throughout, we let $q_t \triangleq \mathrm{law}(\sqrt{1 - \sigma_t^2}\bar{X} + \sigma_t\xi)$, so that $q' = q_T$.[2]

Given $t$, let $F_t^* : \mathbb{R}^d \to \mathbb{R}^d$ denote the map

$$F_t^*(y) = y + \nabla \ln q_t(y). \tag{7}$$

When $t$ is clear from context, we will omit the subscript. Note that

$$\nabla \ln q_t(y) = \frac{\mathbb{E}^{t,y} X_t - y}{\sigma_t^2} = \frac{\mu_t(y) - y}{\sigma_t^2}. \tag{8}$$

We denote our estimate for the score function by $F_t(y)$. When it is clear from the context, we drop the time index $t$ in $F_t$ and $F_t^*$. Moreover, we consider the following processes:

- $(y_t)$: the process given by the true probability flow ODE

$$dy_t = F_t^*(y_t) \, dt, \qquad y_0 \sim q_0 \tag{9}$$

---

[2]Note that the convention in some prior works has been to parametrize time such that $q = q_0$, but we eschew this as our choice makes the notation in our proofs cleaner.

- $(\hat{y}_t)$: the process given by our Picard Algorithm 1 for $t \in [t_i, t_{i+1}]$, for all $1 \leq i \leq n$, as

$$\hat{y}_t \triangleq x_{t_i} + \int_0^{t-t_i} \sum_{i=1}^{D} F_{c_i}(X_{:,i})\phi_i(s)\,\mathrm{d}s,$$

where $X_{:,i}$ are defined in Algorithm 1.

- $(\tilde{y}_t)$: the process that, in the interval $[t_i, t_{i+1}]$ is given by the true probability flow ODE starting from the algorithm iterate $\tilde{y}(t_i)$, for all $1 \leq i \leq n-1$, as defined in 2. Namely, $\tilde{y}(0) = y_{t_i}$, and for all $t \in [t_i, t_{i+1}]$:

$$\frac{d}{dt}\tilde{y}(t) = F_t^*(\tilde{y}(t)).$$

In the course of the proofs, we denote the interval $[t_i, t_{i+1}]$ for a fixed $1 \leq i \leq n-1$ by $[t_0, t_0 + h]$. For an arbitrary curve $z(t) : t \in [t_0, t_0 + h] \to \mathbb{R}^d$, we further use the following sup norm in our calculations:

$$\|z\|_{[t_0,t_0+h]} = \sup_{t_0 \leq t \leq t_0+h} \|z(t)\|.$$

## C DERIVATIVE CALCULATIONS

In order to show that the score function $F_t^*(y_t)$ is a low-degree function of time $t$, our approach is to bound higher derivatives of $F_t^*(y_t)$ with respect to time. Having such bound then enables us to prove that $F_t^*(y_t)$ is close to its $k$th order tailor approximation using the Lagrange error bound, as we state below.

**Fact C.1.** *For function $g : [t_0, t_0 + h] \to \mathbb{R}$, define the low degree approximation $P_{\leq k}g(s)$ with degree $k$ as:*

$$P_{\leq k}g(s) = \sum_{r=0}^{k-1} \frac{d^r g(s)}{ds^r}\bigg|_{s=t_0} \frac{(s-t_0)^r}{r!}.$$

*Then, according to the Lagrange error bound, we have*

$$\|g - P_{\leq k}g\|_{[t_0,t_0+h]} \leq \frac{\max_{s \in [t_0,t_0+h]} \left|\frac{d^k g(s)}{ds^k}\right|}{k!} h^k.$$

Next, we aim to prove an efficient dimension-independent bound on the higher derivatives of the score function. We start from the first order derivative.

The main building blocks of the bound in Lemma 7 are Lemmas 15 and 6. Lemma 15 further decompose into Lemmas 16, 17. In Lemmas 16 and 17 we derive the derivative of the posterior mean $\mu_t(y)$ with respect to time $(t)$ and space $(y)$ variables. In Lemma 6 we combine them together.

### C.1 PROOF OF LEMMA 6

We will require the following calculation whose proof is given in Section C.2 below.

**Lemma 15.** *In the notation of Lemma 6, for any $v \in \mathbb{R}^d$,*

$$\partial_t \mu_t(y) = \mu_t(y) + \frac{1 - \sigma_t^2}{2\sigma_t^4}\mathbb{E}^{t,y}\left(\langle y, X_t\rangle + \|X_t\|^2\right)(X_t - X_t') + \frac{4}{\sigma_t^2}\mathbb{E}^{t,y}\left(\langle y, X_t\rangle - \|X_t\|^2\right)(X_t - X_t'),$$

$$D(\mu_t(y))[v] = \frac{1}{\sigma_t^2}\mathbb{E}^{t,y}\left(\langle y - X_t, v\rangle\right)(X_t - X_t'),$$

*where the expectations on the right-hand sides are with respect to i.i.d. replicas $X, X'$ sampled from the posterior distribution given $y$.*

The proof of Lemma 6 follows from Lemma 15, namely combining Lemmas 16 and 17:

*Proof.* We can write

$$\partial_t(y_t + \nabla \ln q_t(y_t)) = y_t + \nabla \ln q_t(y_t) + \partial_t \nabla \ln q_t(y_t)$$

$$= y_t + \nabla \ln q_t(y_t) + \partial_t \nabla \ln q_t(y)\Big|_{y=y_t} + \nabla^2 q_t(y_t)(y_t + \nabla \ln q_t(y_t)).$$

Now using Lemma 17 with $v = y_t + \nabla \ln q_t(y_t)$, as well as (8), we get

$$\nabla^2 q_t(y_t)(y_t + \nabla \ln q_t(y_t)) = \frac{1}{\sigma_t^2} \nabla \left( \frac{\mathbb{E}^{t,y}X_t - y}{\sigma_t^2} \right)\Big|_{y=y_t} [y_t + \nabla \ln q_t(y_t)]$$

$$= \frac{1}{\sigma_t^2} \left( -y_t + \frac{y_t - \mathbb{E}^{t,y_t}X_t}{\sigma_t^2} + \mathbb{E}^{t,y_t} \frac{1}{\sigma_t^2} \left( \langle y_t - X_t, y_t + \nabla \ln q_t(y_t) \rangle - \|y_t\|^2 + \frac{\|y_t\|^2}{\sigma_t^2} \right)(X_t - X_t') \right)$$

$$= \frac{1}{\sigma_t^2}(\frac{1}{\sigma_t^2} - 1)y_t - \frac{1}{\sigma_t^4}\mathbb{E}^{t,y_t}X_t + \frac{1}{\sigma_t^4}\mathbb{E}^{t,y_t}\left( 2\left\langle y_t, \frac{X_t''}{\sigma_t^2} \right\rangle - \langle X_t, y_t \rangle - \frac{1}{\sigma_t^2}\langle X_t, X_t'' \rangle \right)(X_t - X_t').$$

Combining this with Lemma 17, we obtain

$$\partial_t(y_t + \nabla \ln q_t(y_t)) = \partial_t y_t + \partial_t(\nabla \ln q_t(y))\Big|_{y=y_t} + \nabla^2 q_t(y_t)(y_t + \nabla \ln q_t(y_t))$$

$$= \partial_t y_t + \frac{1}{\sigma_t^2}\partial_t(\mathbb{E}^{t,y}X_t)\Big|_{y=y_t} + \nabla^2 q_t(y_t)(y_t + \nabla \ln q_t(y_t))$$

$$= \left( y_t + \frac{\mathbb{E}^{t,y_t}X_t - y_t}{\sigma_t^2} \right)$$

$$+ \frac{1}{\sigma_t^2}\mathbb{E}^{t,y}X_t + \frac{1 - \sigma_t^2}{2\sigma_t^6}\mathbb{E}^{t,y}\left( \langle y, X_t \rangle + \|X_t\|^2 \right)(X_t - X_t') + \frac{4}{\sigma_t^4}\mathbb{E}^{t,y}\left( \langle y, X_t \rangle - \|X_t\|^2 \right)(X_t - X_t')$$

$$+ \frac{1}{\sigma_t^2}(\frac{1}{\sigma_t^2} - 1)y_t - \frac{1}{\sigma_t^4}\mathbb{E}^{t,y_t}X_t + \frac{1}{\sigma_t^4}\mathbb{E}^{t,y_t}\left( 2\left\langle y_t, \frac{X_t''}{\sigma_t^2} \right\rangle - \langle X_t, y_t \rangle - \frac{1}{\sigma_t^2}\langle X_t, X_t'' \rangle \right)(X_t - X_t').$$

$$(10)$$

where the last term is the result of taking derivative with respect to $\sigma_t$ in the denominator. $\quad\square$

## C.2 Proof of Lemma 15

**Lemma 16.** *Given any time $t$ and $y \in \mathbb{R}^d$, the time derivative of the expectation of the posterior distribution $q^{t,y}$ can be calculated as*

$$\partial_t \mathbb{E}^{t,y}X_t = \mathbb{E}^{t,y}X_t + \frac{1 - \sigma_t^2}{2\sigma_t^4}\mathbb{E}^{t,y}\left( \langle y, X_t \rangle + \|X_t\|^2 \right)(X_t - X_t') + \frac{4}{\sigma_t^2}\mathbb{E}^{t,y}\left( \langle y, X_t \rangle - \|X_t\|^2 \right)(X_t - X_t'),$$

*where $X, X'$ are independent draws from the posterior distribution $q^{t,y}$.*

*Proof.* We have

$$\mathbb{E}^{t,y}X_t = \frac{\int e^{-\frac{\|y - x_t\|^2}{2\sigma_t^2}} x_t q'(\tilde{x}) d\tilde{x}}{\int e^{-\frac{\|y - x_t\|^2}{2\sigma_t^2}} q'(\tilde{x}) d\tilde{x}}$$

with random variables $\xi \sim N(0, \sigma_t^2 I)$, $X_t = \sqrt{1 - \sigma_t^2}\bar{X}_t$, and $\bar{X} \sim q_0$, where $y =$ and we denote the values of $\bar{X}$ and $X_t$ by $\bar{x}$ and $x_t = \sqrt{1 - \sigma_t^2}\bar{x}$, respectively. Then

$$\partial_t \mathbb{E}_{p(\cdot|y), y = \sqrt{1 - \sigma_t^2}\bar{X} + \xi} X_t$$

$$= \frac{\int e^{-\frac{\|y - x_t\|^2}{2\sigma_t^2}} x_t p_0(\bar{x}) d\bar{x}}{\int e^{-\frac{\|y - x_t\|^2}{2\sigma_t^2}} p_0(\bar{x}) d\bar{x}}$$

$$+ 4\frac{\int e^{-\frac{\|y - x_t\|^2}{2\sigma_t^2}} x_t \frac{\langle y - x_t, x_t \rangle}{2\sigma_t^2} p_0(\bar{x}) d\bar{x}}{\int e^{-\frac{\|y - x_t\|^2}{\sigma_t^2}} p_0(\bar{x}) d\bar{x}} - 4\frac{\int e^{-\frac{\|y - x_t\|^2}{2\sigma_t^2}} x_t p_0(\bar{x}) d\bar{x}}{\int e^{-\frac{\|y - x_t\|^2}{2\sigma_t^2}} p_0(\bar{x}) d\bar{x}}\frac{\int e^{-\frac{\|y - x_t\|^2}{2\sigma_t^2}} \frac{\langle y - x_t, x_t \rangle}{2\sigma_t^2} p_0(\bar{x}) d\bar{x}}{\int e^{-\frac{\|y - x_t\|^2}{\sigma_t^2}} p_0(\bar{x}) d\bar{x}}$$

$$+ \frac{\int e^{-\frac{\|y-x_t\|^2}{2\sigma_t^2}} x_t \frac{\|y-x_t\|^2}{2\sigma_t^4}(1-\sigma_t^2)p_0(\bar{x})d\bar{x}}{\int e^{-\frac{\|y-x_t\|^2}{2\sigma_t^2}} p_0(\bar{x})d\bar{x}} - \frac{\int e^{-\frac{\|y-x_t\|^2}{2\sigma_t^2}} x_t p_0(\bar{x})d\bar{x}}{\int e^{-\frac{\|y-x_t\|^2}{2\sigma_t^2}} p_0(\bar{x})d\bar{x}} \frac{\int e^{-\frac{\|y-x_t\|^2}{2\sigma_t^2}} \frac{\|y-x_t\|^2}{2\sigma_t^4}(1-\sigma_t^2)p_0(\bar{x})d\bar{x}}{\int e^{-\frac{\|y-x_t\|^2}{2\sigma_t^2}} p_0(\bar{x})d\bar{x}},$$

where the first line follows from taking $\partial_t$ derivative w.r.t. $x_t$ that is not in the exponent (note that $x_t = \sqrt{1-\sigma_t^2}\bar{x} = e^{-(T-t)}\bar{x}$ is a function of $t$), the second line is the result of taking derivative w.r.t. $x_t$ that is in the exponent, and the final line is taking derivative w.r.t. the $1/\sigma_t^2$ in the exponent. Note that taking derivative w.r.t the numerator and then the denominator in these cases results in a covariance term between $x_t$ and $\langle y - x_t, x_t \rangle$ and $\|y-x_t\|^2$ in the second and third lines, respectively:

$$\partial_t \mathbb{E}^{t,y} X_t = \mathbb{E}^{t,y} X_t + \mathrm{Cov}\left(\frac{(1-\sigma_t^2)\|y-X_t\|^2}{2\sigma_t^4}, X_t\right) + 4\,\mathrm{Cov}\left(\frac{\langle y - X_t, X_t\rangle}{\sigma_t^2}, X_t\right)$$

$$= \mathbb{E}^{t,y} X_t + \frac{1-\sigma_t^2}{2\sigma_t^4}\mathbb{E}^{t,y}\left(\langle y, X_t\rangle + \|X_t\|^2\right)\left(X_t - \mathbb{E}^{t,y} X_t\right) + \frac{4}{\sigma_t^2}\mathbb{E}^{t,y}\left(\langle y, X_t\rangle - \|X_t\|^2\right)\left(X_t - \mathbb{E}^{t,y} X_t\right)$$

$$= \mathbb{E}^{t,y} X_t + \frac{1-\sigma_t^2}{2\sigma_t^4}\mathbb{E}^{t,y}\left(\langle y, X_t\rangle + \|X_t\|^2\right)\left(X_t - X_t'\right) + \frac{4}{\sigma_t^2}\mathbb{E}^{t,y}\left(\langle y, X_t\rangle - \|X_t\|^2\right)\left(X_t - X_t'\right),$$

where $X_t'$ is an independent replica of $X_t$. $\qquad\square$

**Lemma 17.** *The directional derivative of the posterior mean with respect to $y$ is given by*

$$D(\mathbb{E}^{t,y} X_t)[v] = \frac{1}{\sigma_t^2}\mathbb{E}^{t,y}\left(\langle y - X_t, v\rangle\right)\left(X_t - X_t'\right) = \frac{1}{\sigma_t^2}\mathbb{E}^{t,y}\left(\langle y - X_t, v\rangle + \frac{\|y\|^2}{\sigma_t^2} - \|y\|^2\right)\left(X_t - X_t'\right).$$

*Proof.* We have

$$D\left(\mathbb{E}^{t,y} X_t\right)[v] = -\frac{\int e^{-\frac{\|y-x_t\|^2}{2\sigma_t^2}} \left\langle\frac{y-x_t}{\sigma_t^2}, v\right\rangle x_t p_0(\bar{x})d\bar{x}}{\int e^{-\frac{\|y-x_t\|^2}{2\sigma_t^2}} p_0(\bar{x})d\bar{x}} + \frac{\int e^{-\frac{\|y-x_t\|^2}{2\sigma_t^2}} x_t p_0(\bar{x})d\bar{x}}{\int e^{-\frac{\|y-x_t\|^2}{2\sigma_t^2}} p_0(\bar{x})d\bar{x}} \frac{\int e^{-\frac{\|y-x_t\|^2}{2\sigma_t^2}} \left\langle\frac{y-x_t}{\sigma_t^2}, v\right\rangle p_0(\bar{x})d\bar{x}}{\int e^{-\frac{\|y-x_t\|^2}{2\sigma_t^2}} p_0(\bar{x})d\bar{x}}$$

$$= \mathrm{Cov}\left(\left\langle\frac{y-X_t}{\sigma_t^2}, v\right\rangle, X_t\right)$$

$$= \mathbb{E}^{t,y}\frac{1}{\sigma_t^2}\left(\langle y - X_t, v\rangle + \frac{\|y\|^2}{\sigma_t^2} - \|y\|^2\right)\left(X_t - \mathbb{E}^{t,y} X_t\right)$$

$$= \frac{1}{\sigma_t^2}\mathbb{E}^{t,y}\left(\langle y - X_t, v\rangle + \frac{\|y\|^2}{\sigma_t^2} - \|y\|^2\right)\left(X_t - X_t'\right),$$

where we used the fact that $y$ is constant w.r.t the covariance calculation, hence adding $\frac{\|y\|^2}{\sigma_t^2} - \|y\|^2$ does not change the value of the covariance. $\qquad\square$

### C.3 PROOF OF LEMMA 7

In this section, we prove the bound in Lemma 7 on the higher derivatives of the score function.

**Lemma 18** (Restatement of Lemma 7). *The $k$th derivative of the backward ODE vector field can be expanded as*

$$\partial_t^k \left(y_t + \nabla \ln q_t(y_t)\right) = \sum_{r_1,r_2,r_3,r_4 \leq k,\, \mathbf{i}\in[k]^{r_3}, \mathbf{k}\in[k]^{r_4}, \mathbf{l}\in[k]^{r_4}} a_{\mathbf{i},\mathbf{k},\mathbf{l}}\frac{\left(1-\sigma_t^2\right)^{r_1}}{\sigma_t^{r_2}}\mathbb{E}^{t,y_t}\prod_{j=1}^{r_3}\left\langle y_t, X_t^{(\mathbf{i}_j)}\right\rangle\prod_{j=1}^{r_4}\left\langle X_t^{(\mathbf{k}_j)}, X_t^{(\mathbf{l}_j)}\right\rangle X_t,$$

$$\tag{11}$$

$$+ \sum_{r_1,r_2,r_3,r_4 \leq k,\, \mathbf{i}\in[k]^{r_3}, \mathbf{k}\in[k]^{r_4}, \mathbf{l}\in[k]^{r_4}} b_{\mathbf{i},\mathbf{k},\mathbf{l}}\frac{\left(1-\sigma_t^2\right)^{r_1}}{\sigma_t^{r_2}}\mathbb{E}^{t,y_t}\prod_{j=1}^{r_3}\left\langle y_t, X_t^{(\mathbf{i}_j)}\right\rangle\prod_{j=1}^{r_4}\left\langle X_t^{(\mathbf{k}_j)}, X_t^{(\mathbf{l}_j)}\right\rangle y_t,$$

$$\tag{12}$$

*where $r_1 \leq k, r_2 \leq 4k$ and*

$$\sum_{\mathbf{i} \in [k]^{r_3}, \mathbf{k} \in [k]^{r_4}, \mathbf{l} \in [k]^{r_4}} \left| a_{\mathbf{i}, \mathbf{k}, \mathbf{l}} \right| + \left| b_{\mathbf{i}, \mathbf{k}, \mathbf{l}} \right| \leq (74k)^k .$$

*Here the random variables $X^{(i)}$ are independent samples from the posterior distribution $q^{t,y}$, and $\mathbf{i}, \mathbf{k}, \mathbf{l}$ are arbitrary tuples of indices.*

*Proof.* To control the higher derivatives $\partial_t^k (y_t + \nabla \ln q_t(y_t))$, Lemma 6 motivates us to analyze what happens after taking derivative from terms of the form

$$\frac{(1 - \sigma_t^2)^{r_1}}{\sigma_t^{r_2}} \mathbb{E}^{t,y_t} \prod_{j=1}^{r_3} \left\langle y_t, X_t^{(\mathbf{i}_j)} \right\rangle \prod_{j=1}^{r_4} \left\langle X_t^{(\mathbf{k}_j)}, X_t^{(\mathbf{l}_j)} \right\rangle X_t, \tag{13}$$

$$\frac{(1 - \sigma_t^2)^{r_1}}{\sigma_t^{r_2}} \mathbb{E}^{t,y_t} \prod_{j=1}^{r_3} \left\langle y_t, X_t^{(\mathbf{i}_j)} \right\rangle \prod_{j=1}^{r_4} \left\langle X_t^{(\mathbf{k}_j)}, X_t^{(\mathbf{l}_j)} \right\rangle y_t, \tag{14}$$

for some integers $r_1, r_2, r_3, r_4, r \leq k$; such terms naturally appear after taking $k$ derivatives. In particular, we have

$$\partial_t \mathbb{E}^{t,y_t} X_t = \partial_t \mathbb{E}^{t,y} X_t \Big|_{y=y_t} + D(\mathbb{E}^{t,y} X_t)[y_t + \nabla \ln q_t(y_t)]$$

$$= \mathbb{E}^{t,y} X_t + \frac{1 - \sigma_t^2}{2\sigma_t^4} \mathbb{E}^{t,y} \left( \langle y, X_t \rangle + \|X_t\|^2 \right) (X_t - X_t') + \frac{4}{\sigma_t^2} \mathbb{E}^{t,y} \left( \langle y, X_t \rangle - \|X_t\|^2 \right) (X_t - X_t')$$

$$+ \frac{1}{\sigma_t^2} \left( \frac{1}{\sigma_t^2} - 1 \right) y_t - \frac{1}{\sigma_t^4} \mathbb{E}^{t,y_t} X_t + \frac{1}{\sigma_t^4} \mathbb{E}^{t,y_t} \left( 2 \left\langle y_t, \frac{X_t''}{\sigma_t^2} \right\rangle - \langle X_t, y_t \rangle - \frac{1}{\sigma_t^2} \langle X_t, X_t'' \rangle \right) (X_t - X_t'),$$

and

$$\partial_t(y_t) = y_t + \nabla \ln q_t(y_t) = y_t + \frac{\mathbb{E}^{t,y_t} X_t - y_t}{\sigma_t^2}.$$

The key here is to keep track of the total number of $X_t$'s and $y_t$'s that appear in Equation (14). Let's denote this number by $\kappa = 2(r_3 + r_4)$. Then, first note that each step of taking derivatives, we are either taking derivative of some $y_t$ or some $X_t$ in Equation (14). In either case, the total number of $X_t$ and $y_t$'s increase by at most 2, and the power of $\sigma_t^2$ in the denominator, i.e. $r_2$, increases by at most 4. Hence, $\kappa \leq 2k$. Taking derivative wrt $\sigma_t$:

$$\partial_t \frac{(1 - \sigma_t^2)^{r_1}}{\sigma_t^{r_2}} = -r_2 \frac{(1 - \sigma_t^2)^{r_1 + 1}}{\sigma_t^{r_2 + 2}} + \frac{-2r_1 (1 - \sigma_t^2)^{r_1 - 1}}{\sigma_t^{r_2 - 1}}, \tag{15}$$

which increases $r_1$ by at most one and $r_2$ by at most two. Therefore, $r_1 \leq k$ and $r_2 \leq 4k$. On the other hand, taking derivative with respect to either $y_t$ or $X_t$ creates at most $34 = 1 + 2 + 4 \times 2 \times 2 + 2 \times 2 + 2 + 1 + 4 \times 2$ new terms (counting term with coefficient 4 four times) and taking derivative wrt $\sigma_t$ (according to Equation (15)) creates at most $r_2 + 2r_1 \leq 6k$, which means overall we have at most $(2 \times 34 + 6)k = 74k$ new terms generated after taking $k$ derivatives. Therefore, overall, after taking $k$ derivatives we have at most $(74k)^k$ terms generated. □

## C.4  PROOF OF LEMMA 8

Recall the notation $\mathbb{E}^{t,y}$ from Section B. When $t, y$ are clear from context, we will omit the superscript and refer to this expectation by $\mathbb{E}$.

We formally state and prove our upper bound on the derivatives $\partial_t^k (y_t + \nabla \ln q_t(y_t))$ in Lemma 8.

**Lemma 8.** *If $q$ satisfies Assumption 1, $(y_t)_t$ is a solution to (1), and $y_0 \sim q_T$, then the $k^{th}$ derivative of the vector field along the probability flow ODE is bounded as*

$$\left\| \partial_t^k (y_t + \nabla \log q_t(y_t)) \right\|_{p,\infty} \lesssim R \left( \frac{74k}{\sigma_t^2} \right)^k \left( \frac{R}{\sigma_t} + (kp)^{1/2} \right)^{2k}$$

*Proof.* The proof directly follows from Lemmas 7 and 19. □

In Lemma 19 we bound the $p$ norm of the terms that appear in Lemma 7.

**Lemma 19.** *We have*

$$\left\| \frac{(1-\sigma_t^2)^{r_1}}{\sigma_t^{r_2}} \prod_{j=1}^{r_3} \left\langle y_t, X_t^{(\mathbf{i}_j)} \right\rangle \prod_{j=1}^{r_4} \left\langle X_t^{(\mathbf{k}_j)}, X_t^{(\mathbf{l}_j)} \right\rangle X_t \right\|_{p,\infty} \vee \left\| \frac{(1-\sigma_t^2)^{r_1}}{\sigma_t^{r_2}} \prod_{j=1}^{r_3} \left\langle y, X_t^{(\mathbf{i}_j)} \right\rangle \prod_{j=1}^{r_4} \left\langle X_t^{(\mathbf{k}_j)}, X_t^{(\mathbf{l}_j)} \right\rangle y_t \right\|_{p,\infty}$$

$$\leq R \left( \frac{1}{\sigma_t^2} \right)^k \left( \frac{R}{\sigma_t} + (kp)^{1/2} \right)^{2k},$$

*where by $\|v\|_{p,\infty}$ we mean the infinity norm of the $p$th moment of vector $v$.*

*Proof.* First, from the fact that the radius of support of $q_0$ is bounded by $R$, we have

$$\left\| \frac{(1-\sigma_t^2)^{r_1}}{\sigma_t^{r_2}} \prod_{j=1}^{r_3} \left\langle y_t, X_t^{(\mathbf{i}_j)} \right\rangle \prod_{j=1}^{r_4} \left\langle X_t^{(\mathbf{k}_j)}, X_t^{(\mathbf{l}_j)} \right\rangle X_t \right\|_{p,\infty} \leq \frac{1}{\sigma_t^{r_2}} \left\| \prod_{j=1}^{r_3} \left\langle y_t, X_t^{(\mathbf{i}_j)} \right\rangle \right\|_p R^{2r_4+1}.$$

Now using the derivation in Lemma 4.4 in Gatmiry et al. (2024), we get

$$\left\| \prod_{j=1}^{r_3} \left\langle y_t, X_t^{(\mathbf{i}_j)} \right\rangle \right\|_p \leq R^{2r_3} \left( 1 + \frac{(r_3 p)^{1/2} \sigma_t}{R} \right)^{r_3}.$$

Plugging this above and using $R \geq 1$:

$$LHS \leq \frac{R^{2r_3+2r_4+1}}{\sigma_t^{r_2}} \left( 1 + \frac{(r_3 p)^{1/2} \sigma_t}{R} \right)^{r_3} \leq R \left( \frac{R}{\sigma_t^2} \right)^{2k} \left( 1 + \frac{(kp)^{1/2} \sigma_t}{R} \right)^k$$

$$\leq R \left( \frac{1}{\sigma_t} \right)^{2k} \left( \frac{R}{\sigma_t} \right)^{2k} \left( 1 + \frac{(kp)^{1/2} \sigma_t}{R} \right)^{2k}$$

$$\leq R \frac{1}{\sigma_t^{2k}} \left( \frac{R}{\sigma_t} + (kp)^{1/2} \right)^{2k}$$

Similarly for the other term, using Cauchy-Schwarz

$$\left\| \frac{(1-\sigma_t^2)^{r_1}}{\sigma_t^{r_2}} \prod_{j=1}^{r_3} \left\langle y, X_t^{(\mathbf{i}_j)} \right\rangle \prod_{j=1}^{r_4} \left\langle X_t^{(\mathbf{k}_j)}, X_t^{(\mathbf{l}_j)} \right\rangle y \right\|_{p,\infty} \leq \frac{1}{\sigma_t^{r_2}} \left\| \prod_{j=1}^{r_3} \left\langle y, X_t^{(\mathbf{i}_j)} \right\rangle \right\|_{2p} R^{2r_4} \|y\|_{2,\infty}$$

$$\leq \frac{1}{\sigma_t^{r_2-1}} R^{2r_3} \left( 1 + \frac{(2r_3 p)^{1/2} \sigma_t}{R} \right)^{r_3} R^{2r_4}$$

$$\leq \frac{1}{\sigma_t^{2k}} \left( \frac{R}{\sigma_t} + (kp)^{1/2} \right)^{2k}. \qquad \square$$

Next, based on Lemmas 7 and 19, we state our final estimate on $\left\| \partial_t^k \left( y_t + \nabla \ln q_t(y_t) \right) \right\|_{\infty,p}$.

## D    PROOF OF LEMMA 9

We prove Theorem 10 by translating the inequality in Lemma 8 from $y$ to a close by $\tilde{y}$ via a coupling between the distribution of $X_t$ conditioned on $y$ and $\tilde{y}$. Note that the argument of Lemma 8 is not immediate for $\tilde{y}$ because it does not follow the distribution $q * N(0, \sigma_t^2 I)$ as $y$ does. In particular, recall that $\bar{X} \sim q'$ and $\xi \sim N(0, \sigma_t^2 I)$ and $y_t = X_t + \xi$, $X_t = \sqrt{1-\sigma_t^2} \bar{X}$. Now for every $\varepsilon_1 > 0$, consider the event

$$\mathcal{A}_1 := \left\{ \|y_t - X_t\| \leq \sqrt{d} + \sqrt{\ln(1/\varepsilon_1)} \right\}, \tag{16}$$

and note that $\mathbb{P}(\mathcal{A}_1) \geq 1 - \varepsilon_1$, since $\sigma_t \leq 1$.

**Lemma 9** (Coupling). *Let $\tilde{y}$ a random variable such that $\|y - \tilde{y}\| \leq \delta$ for $\delta \leq \frac{1}{6(\sqrt{d}+\sqrt{\ln(1/\varepsilon_1)})}$.*
*Now consider the conditional distribution $p(.|\tilde{y})$, where $p(x, y)$ is the joint distribution of $(X_t, y_t)$,*
*for $y_t = X_t + \xi$, $X_t = \sqrt{1 - \sigma_t^2}\bar{X}$, $\bar{X} \sim q'$, and $\xi \sim N(0, \sigma_t^2 I)$. Then, given $T - t \geq 1$, under the*
*event $\mathcal{A}_1$ defined in Equation (16), we have*

$$TV(p(.|y), p(.|\tilde{y})) \leq 8\delta(\sqrt{d} + \sqrt{\ln(1/\varepsilon_1)}) \tag{5}$$

*In particular, Inequality (5) holds with probability at least $1 - \varepsilon_1$.*

*Proof.* Since $T - t \geq 1$, we have

$$\sigma_t^2 = 1 - e^{-2(T-t)} \geq 1 - e^{-2} \geq 0.5.$$

Hence, we have

$$\left| \ln\left( \frac{p(y|x)}{p(\tilde{y}|x)} \right) \right| = \frac{1}{2\sigma_t^2} \left| \|y - x\|^2 - \|\tilde{y} - x\|^2 \right|$$

$$\leq \left| \|y - x\|^2 - \|\tilde{y} - x\|^2 \right|$$

$$\leq 2\|y - \tilde{y}\| \|y - x\| + \|y - \tilde{y}\|^2$$

$$\leq 2\delta(\sqrt{d} + \sqrt{\ln(1/\varepsilon_1)}) + \delta^2$$

$$\leq 3\delta(\sqrt{d} + \sqrt{\ln(1/\varepsilon_1)}),$$

and by the assumed bound on $\delta$, we have $2\delta(\sqrt{d} + \sqrt{\ln(1/\varepsilon_1)}) + \delta^2 \leq 0.5$, which implies

$$(1 - 3\delta(\sqrt{d} + \sqrt{\ln(1/\varepsilon_1)}))p(y|x) \leq p(\tilde{y}|x) \leq p(y|x)(1 + 3\delta(\sqrt{d} + \sqrt{\ln(1/\varepsilon_1)})).$$

Multiplying $p(x)$ on both sides, we get

$$(1 - 3\delta(\sqrt{d} + \sqrt{\ln(1/\varepsilon_1)}))p(y|x)p(x) \leq p(\tilde{y}|x)p(x) \leq p(y|x)p(x)(1 + 3\delta(\sqrt{d} + \sqrt{\ln(1/\varepsilon_1)})).$$

But from Bayes' rule, it is easy to see that this implies

$$(1 - 3\delta(\sqrt{d} + \sqrt{\ln(1/\varepsilon_1)}))^2 p(x|y) \leq p(x|\tilde{y}) \leq p(x|y)(1 + 3\delta(\sqrt{d} + \sqrt{\ln(1/\varepsilon_1)}))^2.$$

Given that $(1 + 3\delta(\sqrt{d} + \sqrt{\ln(1/\varepsilon_1)}))^2 \leq 1 + 8\delta(\sqrt{d} + \sqrt{\ln(1/\varepsilon_1)})$ by the assumed bound on $\delta$, we conclude that

$$TV(p(.|y), p(.|\tilde{y})) \leq 8\delta(\sqrt{d} + \sqrt{\ln(1/\varepsilon_1)}). \qquad \square$$

# E  PROOF OF THEOREM 10

Here we show a similar bound on the derivatives of the vector field for the curve $\tilde{y}$ instead of $y$, where $y$ and $\tilde{y}$ are defined in Section B. Throughout this section, we assume $\tilde{y}_t$ and $y_t$ are close by $\delta$ in the window $[t_0, t_0 + h]$:

$$\|y_t - \tilde{y}_t\|_{[t_0, t_0+h]} \leq \delta. \tag{17}$$

Later on in the proof of Theorem 34, we will show by induction that it is valid to assume Equation (17) holds. When it is clear, we will drop the $t$ indices from the variables for clarity. First, we restate Theorem 10 and show the short proof based on the rest of this section. We then get into the details of proving the intermediate lemmas.

**Theorem 10.** *Let $\delta \in (0, 1)$. If $q$ satisfies Assumption 1, and $(\tilde{y}_t)_t$ is given by initializing the reverse process at a distribution which is $\delta$ $W_2$-**close** to $q_0$ and running the probability flow ODE. Then, for $\delta$ small enough (as rigorously characterized in Lemma 25), with probability at least $1 - \delta$ over the randomness of the initialization, the $k^{th}$ derivative of the vector field along the probability flow ODE is bounded at $\tilde{y}_t$ by*

$$\left\| \partial_t^k \left( \tilde{y}_t + \nabla \log q_t(\tilde{y}_t) \right) \right\|_\infty \lesssim R \left( \frac{k}{\sigma_t^2} \right)^k \left( \frac{R}{\sigma_t} \right)^{2k} + (k \log(74kd/\delta))^{2k}.$$

*Proof.* For simplicity of exposition we will drop the time index $t$ from $y_t, X_t$ in the rest of this section, and write $y = X + \xi$, where $\bar{X} \sim q'$, $X = \sqrt{1 - \sigma_t^2}\bar{X}$ and $\xi \sim N(0, \sigma_t^2 I)$. Let $p(.|y)$ be the distribution of $X$ conditioned on $y$ and let $\tilde{X}$ be a sample from the conditional $p(.|\tilde{y})$. For index $i$ let $X^{(i)}$ and $\tilde{X}^{(i)}$ be an independent and identical sample of $X$ and $\tilde{X}$, respectively. Now using Lemma 25 with $\delta_1 = \frac{\delta_2}{d(60k)^k}$, we get

$$\mathbb{P}\left(\left|\mathbb{E}_{\{\tilde{X}^{(i)}\}}\frac{(1-\sigma_t^2)^{r_1}}{\sigma_t^{r_2}}\prod_{j=1}^{r_3}\left\langle\tilde{y},\tilde{X}^{(\mathbf{i}_j)}\right\rangle\prod_{j=1}^{r_4}\left\langle\tilde{X}^{(\mathbf{k}_j)},\tilde{X}^{(\mathbf{l}_j)}\right\rangle X\right|_\infty \geq eR\left(\frac{1}{\sigma_t^2}\right)^k\left(\frac{R}{\sigma_t}\right)^{2k} + e\left(k\log(74kd/\delta_2)\right)^k\right)$$

$$\leq \delta_2.$$

Taking union bound over all the terms, we get with probability at least $1 - \delta_2$

$$\left\|\partial_t^k(\tilde{y}_t + \nabla\log q_t(\tilde{y}_t))\right\|_\infty \leq e\left(R + \delta + \sigma_t + 1\right)\left(\frac{kh}{\sigma_t^2}\right)^k\left(\frac{R}{\sigma_t}\right)^{2k} + e\left(k\log(74kd/\delta_2)\right)^{2k}$$

$\square$

Next, we derive a pessimistic bound on $\mathbb{E}\left|\prod_{j=1}^{r_3}\left\langle\tilde{y}, X^{(\mathbf{i}_j)}\right\rangle\right|$ for all ordered tuples of indices $\mathbf{i}$, where each $X^{(i)}$ denotes an independent sample from the posterior distribution $p(\cdot \mid y)$, by obtaining a pessimistic bound on the term $\left\|\prod_{j=1}^{r_3}\left\langle y, X^{(\mathbf{i}_j)}\right\rangle\right\|_p$

**Lemma 20.** *For all tuples of indices $\mathbf{i}$, we have*

$$\mathbb{E}\left|\prod_{j=1}^{r_3}\left\langle\tilde{y}, X^{(\mathbf{i}_j)}\right\rangle\right| \leq \left(R^2\left(1 + \frac{(r_3p)^{1/2}\sigma_t}{R}\right) + \delta R\right)^{r_3}$$

*Proof.* Using Cauchy-Schwarz,

$$\left\|\prod_{j=1}^{r_3}\left\langle\tilde{y}, X^{(\mathbf{i}_j)}\right\rangle\right\|_p \leq \sum_{S\subseteq[k]}\left\|\prod_{j\in S}\left\langle y, X^{(\mathbf{i}_j)}\right\rangle\right\|_p\left\|\prod_{j\notin S}\left\langle\tilde{y}-y, X^{(\mathbf{i}_j)}\right\rangle\right\|_p$$

$$\leq \sum_{\ell=1}^{r_3}\binom{r_3}{\ell}\left(R^2\left(1 + \frac{(r_3p)^{1/2}\sigma_t}{R}\right)\right)^\ell(\delta R)^{r_3-\ell}$$

$$\leq \left(R^2\left(1 + \frac{(r_3p)^{1/2}\sigma_t}{R}\right) + \delta R\right)^{r_3}.$$

$\square$

**Lemma 21.** *For $\nu \triangleq \left(3\delta(\sqrt{d} + \sqrt{\ln(1/\varepsilon_1)})\right)^{1/p}$, we have for all tuples of indices $\mathbf{i}$ that*

$$\left\|\mathbf{1}\left\{\mathcal{A}_1\right\}\prod_{j=1}^{r_3}\left\langle\tilde{y}, \tilde{X}^{(\mathbf{i}_j)}\right\rangle\right\|_p \leq \left(R^2\left(1 + \frac{(2r_3p)^{1/2}\sigma_t}{R}\right) + \delta R + \nu R\left(\sqrt{2pd} + R + \delta\right)\right)^{r_3}. \tag{18}$$

*Proof.* First, note that we have the following bound:

$$\left\|\prod_{j=1}^{r_3}\left\langle\tilde{y}, \tilde{X}^{(\mathbf{i}_j)}\right\rangle\right\|_p^p = \mathbb{E}\prod_{j=1}^{r_3}\left\langle\tilde{y}, \tilde{X}^{(\mathbf{i}_j)}\right\rangle^p \tag{19}$$

$$\leq \mathbb{E}\prod_{j=1}^{r_3}\|\tilde{y}\|^p\left\|\tilde{X}^{(\mathbf{i}_j)}\right\|^p \tag{20}$$

$$\leq \mathbb{E}\prod_{j=1}^{r_3}\left(\|y - \tilde{y}\| + \|\xi\| + \|X^{\mathbf{i}_j}\|\right)^p\left\|\tilde{X}^{(\mathbf{i}_j)}\right\|^p \tag{21}$$

$$\leq \mathbb{E} \prod_{j=1}^{r_3} \left( \delta + \|\xi\| + R \right)^p R^p \tag{22}$$

$$\leq \mathbb{E} \sum_{\ell=1}^{r_3 p} \binom{r_3 p}{\ell} \|\xi\|^\ell \left( R + \delta \right)^{r_3 p - \ell} R^{r_3 p} \tag{23}$$

$$\leq \sum_{\ell=1}^{r_3 p} \binom{r_3 p}{\ell} \left( \sqrt{\ell d} \right)^\ell \left( R + \delta \right)^{r_3 p - \ell} R^{r_3 p} \tag{24}$$

$$= \left( R \left( \sqrt{pd} + R + \delta \right) \right)^{r_3 p}, \tag{25}$$

which implies

$$\left\| \prod_{j=1}^{r_3} \left\langle \tilde{y}, \tilde{X}^{(\mathbf{i}_j)} \right\rangle \right\|_p \leq \left( R \left( \sqrt{pd} + R + \delta \right) \right)^{r_3}.$$

Therefore, applying Lemma **??** with $\varepsilon_1 = \delta$, on event $\mathcal{A}_1$ we have

$$TV(p(.|y), p(.|\tilde{y})) \leq 3\delta(\sqrt{d} + \sqrt{\ln(1/\varepsilon_1)}),$$

which means for all $m$,

$$TV(X^{(m)}, \tilde{X}^{(m)}) \leq 3\delta(\sqrt{d} + \sqrt{\ln(1/\varepsilon_1)}).$$

Therefore, the optimal coupling between $X^{(m)}$ and $\tilde{X}^{(m)}$ conditioned on $y$ satisfies

$$\mathbb{P}\left( X^{(m)} = \tilde{X}^{(m)} \right) \geq 1 - 3\delta(\sqrt{d} + \sqrt{\ln(1/\varepsilon_1)}) \tag{26}$$

Now equipped with this coupling, for a subset $\mathcal{U} \subset [r_3]$, let $\mathcal{A}_2 = \mathcal{A}_2(\mathcal{U})$ be the event that $X^{(\mathbf{i}_j)} = \tilde{X}^{(\mathbf{i}_j)}$ for all $j \in \mathcal{U}$. Then we can write

$$\mathbb{E} \left( 1\left\{ \mathcal{A}_1 \cap \mathcal{A}_2(\mathcal{U}) \right\} \prod_{j=1}^{r_3} \left\langle \tilde{y}, \tilde{X}^{(\mathbf{i}_j)} \right\rangle \right)^p$$

$$= \mathbb{E} \left( 1\left\{ \mathcal{A}_1 \cap \mathcal{A}_2(\mathcal{U}) \right\} \prod_{j \in \mathcal{U}} \left\langle \tilde{y}, \tilde{X}^{(\mathbf{i}_j)} \right\rangle \right)^p \left( 1\left\{ \mathcal{A}_1 \cap \mathcal{A}_2(\mathcal{U}) \right\} \prod_{j \notin \mathcal{U}} \left\langle \tilde{y}, \tilde{X}^{(\mathbf{i}_j)} \right\rangle \right)^p.$$

$$\leq \sqrt{ \mathbb{E} \left( 1\left\{ \mathcal{A}_1 \cap \mathcal{A}_2(\mathcal{U}) \right\} \prod_{j \in \mathcal{U}} \left\langle \tilde{y}, \tilde{X}^{(\mathbf{i}_j)} \right\rangle \right)^{2p}} \sqrt{ \mathbb{E} \left( 1\left\{ \mathcal{A}_1 \cap \mathcal{A}_2(\mathcal{U}) \right\} \prod_{j \notin \mathcal{U}} \left\langle \tilde{y}, \tilde{X}^{(\mathbf{i}_j)} \right\rangle \right)^{2p}} \tag{27}$$

But on one hand, from Equation (26) and the independence of the couplings, we have $\mathbb{P}(\mathcal{A}_2(\mathcal{U})|\mathcal{A}_1) \leq \left( 3\delta(\sqrt{d} + \sqrt{\ln(1/\varepsilon_1)}) \right)^{r_3 - |\mathcal{U}|}$. Therefore, similar to the derivation in (25)

$$\mathbb{E} \left( 1\left\{ \mathcal{A}_1 \cap \mathcal{A}_2(\mathcal{U}) \right\} \prod_{j \notin \mathcal{U}} \left\langle \tilde{y}, \tilde{X}^{(\mathbf{i}_j)} \right\rangle \right)^p$$

$$\leq \mathbb{E} 1\left\{ \mathcal{A}_1 \right\} \sum_{\ell=1}^{(r_3 - |\mathcal{U}|)p} \binom{(r_3 - |\mathcal{U}|)p}{\ell} \|\xi\|^\ell \left( R + \delta \right)^{(r_3 - |\mathcal{U}|)p - \ell} R^{(r_3 - |\mathcal{U}|)p} \mathbb{P}_{|y}(\mathcal{A}_2(\mathcal{U})|\mathcal{A}_1)$$

$$\leq \mathbb{P}_{|y}(\mathcal{A}_2(\mathcal{U})|\mathcal{A}_1) \sum_{\ell=1}^{(r_3 - |\mathcal{U}|)p} \binom{(r_3 - |\mathcal{U}|)p}{\ell} \|\xi\|^\ell \left( R + \delta \right)^{(r_3 - |\mathcal{U}|)p - \ell} R^{(r_3 - |\mathcal{U}|)p}$$

$$\leq \left( 3\delta(\sqrt{d} + \sqrt{\ln(1/\varepsilon_1)}) \right)^{|\mathcal{U}|} \left( R \left( \sqrt{pd} + R + \delta \right) \right)^{(r_3 - |\mathcal{U}|)p}$$

$$= \nu^{p(r_3 - |\mathcal{U}|)} \left( R \left( \sqrt{pd} + R + \delta \right) \right)^{(r_3 - |\mathcal{U}|)p},$$

where $\nu \triangleq \left(3\delta(\sqrt{d} + \sqrt{\ln(1/\varepsilon_1)})\right)^{1/p}$. This implies

$$\left\|1\{\mathcal{A}_1 \cap \mathcal{A}_2(\mathcal{U})\} \prod_{j \notin \mathcal{U}}^{r_3} \left\langle \tilde{y}, \tilde{X}^{(\mathbf{i}_j)} \right\rangle\right\|_p \leq \left(3\delta(\sqrt{d} + \sqrt{\ln(1/\varepsilon_1)})\right)^{1/p} \left(R\left(\sqrt{pd} + R + \delta\right)\right)^{r_3 - |\mathcal{U}|}.$$

(28)

For the first term, we can use the coupling and substitute $\tilde{X}^{(\mathbf{i}_j)}$'s by $X^{(\mathbf{i}_j)}$ and then use Lemma 20:

$$\mathbb{E}\left(1\{\mathcal{A}_1 \cap \mathcal{A}_2(\mathcal{U})\} \prod_{j \in \mathcal{U}} \left\langle \tilde{y}, \tilde{X}^{(\mathbf{i}_j)} \right\rangle\right)^p = \mathbb{E}\left(1\{\mathcal{A}_1 \cap \mathcal{A}_2(\mathcal{U})\} \prod_{j \in \mathcal{U}} \left\langle \tilde{y}, X^{(\mathbf{i}_j)} \right\rangle\right)^p \tag{29}$$

$$\leq \mathbb{E}\left(\prod_{j \in \mathcal{U}} \left\langle \tilde{y}, \tilde{X}^{(\mathbf{i}_j)} \right\rangle\right)^p \tag{30}$$

$$\leq \left(R^2\left(1 + \frac{(r_3 p)^{1/2} \sigma_t}{R}\right) + \delta R\right)^{p|\mathcal{U}|}. \tag{31}$$

Combining Equations (31) and (28) and plugging back into Equation (27):

$$\mathbb{E}\left(1\{\mathcal{A}_1\} \prod_{j=1}^{r_3} \left\langle \tilde{y}, \tilde{X}^{(\mathbf{i}_j)} \right\rangle\right)^p$$

$$= \sum_{\mathcal{U} \subseteq [r_3]} \mathbb{E}\left(1\{\mathcal{A}_1 \cap \mathcal{A}_2(\mathcal{U})\} \prod_{j=1}^{r_3} \left\langle \tilde{y}, \tilde{X}^{(\mathbf{i}_j)} \right\rangle\right)^p$$

$$\leq \sum_{\mathcal{U} \subseteq [r_3]} \left(R^2\left(1 + \frac{(2r_3 p)^{1/2} \sigma_t}{R}\right) + \delta R\right)^{p|\mathcal{U}|} \left(\nu R\left(\sqrt{2pd} + R + \delta\right)\right)^{(r_3 - |\mathcal{U}|)p}$$

$$\leq \left(R^2\left(1 + \frac{(2r_3 p)^{1/2} \sigma_t}{R}\right) + \delta R + \nu R\left(\sqrt{2pd} + R + \delta\right)\right)^{r_3 p}. \qquad \square$$

**Lemma 22** (Replica bounds for the coupled variables). *Suppose we have*

$$\delta \leq R \wedge \frac{1}{\left(\sqrt{d} + \sqrt{\ln(1/\varepsilon_1)}\right)\left(\sqrt{2pd}/R + 1\right)^p}.$$

*Then for $\tilde{y}$ as defined in Section B, let $\tilde{X}^{(i)}$ be independent samples from the posterior distribution $p(\cdot \mid \tilde{y})$. Then, for integers $r_2, r_3, r_4$ that satisfy $r_2 \leq 2(r_3 + r_4) \leq 2k$, under event $\mathcal{A}_1$ defined in Section **??**, we have for all tuples of indices $\mathbf{i}, \mathbf{k}, \mathbf{l} \in$,*

$$\left\|1\{\mathcal{A}_1\} \frac{(1 - \sigma_t^2)^{r_1}}{\sigma_t^{r_2}} \prod_{j=1}^{r_3} \left\langle \tilde{y}, \tilde{X}^{(\mathbf{i}_j)} \right\rangle \prod_{j=1}^{r_4} \left\langle \tilde{X}^{(\mathbf{k}_j)}, \tilde{X}^{(\mathbf{l}_j)} \right\rangle \tilde{X}\right\|_{p,\infty} \lesssim \frac{R}{\sigma_t^{2k}} \left(\frac{R}{\sigma_t} + (kp)^{1/2}\right)^{2k},$$

$$\left\|1\{\mathcal{A}_1\} \frac{(1 - \sigma_t^2)^{r_1}}{\sigma_t^{r_2}} \prod_{j=1}^{r_3} \left\langle \tilde{y}, \tilde{X}^{(\mathbf{i}_j)} \right\rangle \prod_{j=1}^{r_4} \left\langle \tilde{X}^{(\mathbf{k}_j)}, \tilde{X}^{(\mathbf{l}_j)} \right\rangle \tilde{y}\right\|_{p,\infty} \lesssim \frac{R+1}{\sigma_t^{2k}} \left(\frac{R}{\sigma_t} + (kp)^{1/2}\right)^{2k},$$

*where by $\|v\|_{p,\infty}$ we mean the infinity norm of the pth moment of the vector $v$.*

*Proof.* Note that given the condition on $\delta$, then in the bound of Equation (18) in Lemma 21 the second term is dominated by the first term, which implies

$$\left\|1\{\mathcal{A}_1\} \prod_{j=1}^{r_3} \left\langle \tilde{y}, \tilde{X}^{(\mathbf{i}_j)} \right\rangle\right\|_p \lesssim \left(R^2\left(1 + \frac{(r_3 p)^{1/2} \sigma_t}{R}\right) + \delta R\right)^{r_3}$$

$$\lesssim \left(R^2\left(1 + \frac{(r_3 p)^{1/2} \sigma_t}{R}\right)\right)^{r_3}.$$

Therefore, from $r_2 \leq 2(r_3 + r_4)$ and using the fact that support of $q_0$ has bounded radius $R$:

$$\left\| 1\{\mathcal{A}_1\} \frac{(1-\sigma_t^2)^{r_1}}{\sigma_t^{r_2}} \prod_{j=1}^{r_3} \left\langle \tilde{y}, \tilde{X}^{(\mathbf{i}_j)} \right\rangle \prod_{j=1}^{r_4} \left\langle \tilde{X}^{(\mathbf{k}_j)}, \tilde{X}^{(\mathbf{l}_j)} \right\rangle \tilde{X} \right\|_{p,\infty}$$

$$\leq \frac{1}{\sigma_t^{r_2}} \left\| 1\{\mathcal{A}_1\} \prod_{j=1}^{r_3} \left\langle \tilde{y}, \tilde{X}^{(\mathbf{i}_j)} \right\rangle \right\|_p R^{2r_4+1}$$

$$\lesssim R \left( \frac{1}{\sigma_t^2} \right)^{r_3+r_4} \left( \frac{R^2}{\sigma_t^2} \right)^{r_4} \left( \frac{R^2}{\sigma_t^2} + \frac{R}{\sigma_t}(r_3 p)^{1/2} \right)^{r_3}$$

$$\leq \frac{R}{\sigma_t^{2k}} \left( \frac{R}{\sigma_t} \right)^{2r_4} \left( \left( \frac{R}{\sigma_t} + (r_3 p)^{1/2} \right)^2 \right)^{r_3}$$

$$\leq \frac{R}{\sigma_t^{2k}} \left( \frac{R}{\sigma_t} + (kp)^{1/2} \right)^{2k}.$$

where we used the fact that $r_3 + r_4 \leq k$. Similarly for the other term, using Cauchy Swartz

$$\left\| 1\{\mathcal{A}_1\} \frac{(1-\sigma_t^2)^{r_1}}{\sigma_t^{r_2}} \prod_{j=1}^{r_3} \left\langle \tilde{y}, \tilde{X}^{(\mathbf{i}_j)} \right\rangle \prod_{j=1}^{r_4} \left\langle \tilde{X}^{(\mathbf{k}_j)}, \tilde{X}^{(\mathbf{l}_j)} \right\rangle y \right\|_{p,\infty}$$

$$\leq \frac{1}{\sigma_t^{r_2}} \left\| \prod_{j=1}^{r_3} \left\langle \tilde{y}, \tilde{X}^{(\mathbf{i}_j)} \right\rangle \right\|_{2p} R^{2r_4} \left( \|\tilde{y} - y\|_{2,\infty} + \|y\|_{2,\infty} \right)$$

$$\leq \frac{1}{\sigma_t^{r_2}} \left\| \prod_{j=1}^{r_3} \left\langle \tilde{y}, \tilde{X}^{(\mathbf{i}_j)} \right\rangle \right\|_{2p} R^{2r_4} \left( R + \sigma_t + \delta \right)$$

$$\lesssim \frac{R+1}{\sigma_t^{2k}} \left( \frac{R}{\sigma_t} + (kp)^{1/2} \right)^{2k}. \qquad \square$$

**Lemma 23** (High probability control on higher derivatives – fixed time). *In the setting of Lemma 22 let*

$$g(t) = \tilde{y}_t + \nabla \log q_t(\tilde{y}_t),$$

*where $\tilde{y}(t)$ is defined in Eq. (7). Then, for constant $c_1$ and arbitrary coordinate $i$,*

$$\mathbb{P} \left( \left| \partial_t^k g_i(t) \right| \geq c_1 (60k)^k (R+1) \left( \frac{1}{\sigma_t^2} \right)^k \left( \frac{R}{\sigma_t} + \sqrt{\ln(1/\varepsilon_1)} \right)^{2k} \right) \leq \varepsilon_1.$$

*Proof.* Using Lemmas 22 and 7, there is a constant $c_1$ such that

$$\left\| \partial_t^k g_i(t) \right\|_p = \left\| \partial_t^k (y_{t,i} + \partial_i \log q_t(y_t)) \right\|_p$$

$$\leq (c_1/e)(60k)^k (R+1) \left( \frac{1}{\sigma_t^2} \right)^k \left( \frac{R}{\sigma_t} + (kp)^{1/2} \right)^{2k}.$$

Moreover, for fixed $j \leq d$, let $\mathcal{A}_3 \subseteq \mathcal{A}_1$ be the event inside $\mathcal{A}_1$ where

$$\left| \partial_t^k g_i(t) \right| \geq c_1 (60k)^k (R+1) \left( \frac{1}{\sigma_t^2} \right)^k \left( \frac{R}{\sigma_t} + \sqrt{\ln(1/\varepsilon_1)} \right)^{2k}.$$

Taking expectation with respect to $y$

$$\mathbb{E}_y \mathbb{E}_{\{\tilde{X}^{(i)}\}} \left| \partial_t^k g_i(t) \right|^p$$

$$\geq \mathbb{P}(\mathcal{A}_3) \left( c_1 (60k)^k (R+1) \left( \frac{1}{\sigma_t^2} \right)^k \left( \frac{R}{\sigma_t} + \sqrt{\ln(1/\varepsilon_1)} \right)^{2k} \right)^p.$$

Now setting $p = \ln(1/\varepsilon_1)/k$, we get

$$\mathbb{P}(\mathcal{A}_3) \leq \left( \frac{\frac{R}{\sigma_t} + (kp)^{1/2}}{e\frac{R}{\sigma_t} + e\sqrt{\ln(1/\varepsilon_1)}} \right)^{2kp} = \left( \frac{1}{e} \right)^{2\ln(1/\varepsilon_1)} \leq \varepsilon_1. \qquad \square$$

**Lemma 24.** *In the same setting of Lemma 23, for*

$$\delta \leq R \wedge \frac{1}{\left( \sqrt{d} + \sqrt{\ln(1/\varepsilon_1)} \right) \left( \sqrt{2\ln(1/\varepsilon)d}/(R\sqrt{k}) + 1 \right)^{\ln(1/\varepsilon_1)/k}},$$

*for some constant $c_1$, we have*

$$\mathbb{P}\left( \sup_{\tilde{y}: \|\tilde{y}-y\| \leq \delta} \left| \partial_t^k g_i(t) \right|_{s=0} \geq c_1(60k)^k(R+1) \left( \frac{1}{\sigma_t^2} \right)^k \left( \frac{R}{\sigma_t} + \sqrt{\ln(1/\varepsilon_1)} \right)^{2k} \right) \leq \varepsilon_1.$$

*Proof.* Note that the $\tilde{y}$ can be an arbitrary mapping of $y$ in the neighborhood $\|y - \tilde{y}\| \leq \delta$. Hence, we pick $\tilde{y}$ to be

$$\tilde{y}(y) \triangleq \text{argmax}_{\tilde{y}: \|\tilde{y}-y\| \leq \delta} \left| \partial_t^k g_i(t) \right|_{t=0}.$$

The result then follows from Lemma 23. $\qquad \square$

**Lemma 25.** *Let the distance between the two curves, namely the true and algorithm curves, be upper bounded by*

$$\tilde{\delta} \triangleq R \wedge \frac{1}{\left( \sqrt{d} + \sqrt{\ln(1/\varepsilon_1)} \right) \left( \sqrt{2\ln(1/\varepsilon_1)d}/(R\sqrt{k}) + 1 \right)^{\ln(1/\varepsilon_1)/k}}, \tag{32}$$

*that is $\sup_{t \in [t_0, t_0+h]} \|y(t) - \tilde{y}(t)\| \leq \tilde{\delta}$, where $\tilde{y}(t)$ is defined in Lemma 24 and $y$ is defined analogously with $y(0) = y$. Then*

$$\mathbb{P}\left( \sup_{t \in [t_0, t_0+h]} \left| \partial_t^k g_i(t) \right|_{t=0} \geq C_1 \right) \leq 2\varepsilon_1.$$

*where*

$$C_1 \triangleq c_1(60k)^k(R+1) \left( \frac{1}{\sigma_t^2} \right)^k \left( \frac{R}{\sigma_t} + \sqrt{\ln\left( \frac{R + \sqrt{d} + \sqrt{\ln(1/\varepsilon_1)}}{\varepsilon_1} \right)} \right)^{2k}.$$

*Proof.* Suppose we take a cover $\mathcal{C}$ of the interval $[t_0, t_0 + h]$ with accuracy $\frac{\tilde{\delta}}{6\left( R + \sqrt{d} + \sqrt{\ln(1/\varepsilon_1)} \right)}$.

Then, using Lemma 24 with a union bound, particularly since $2\tilde{\delta}$ satisfies the condition of this lemma, we get (using $h \leq 1$)

$$\mathbb{P}\left( \sup_{\tilde{y}: \|\tilde{y}-y\| \leq 2\tilde{\delta}, y \in \mathcal{C}} \left| \partial_t^k g_i(t) \right|_{t=0} \geq C_1 \right) \leq \varepsilon_1. \tag{33}$$

On the other hand, for every $s \in [t_0, t_0 + h]$, if we assume $\bar{s} \in \mathcal{C}$ be the closest element of the cover to $s$, then from Lemma 38 and the choice of accuracy of the cover:

$$\|y(\bar{t}) - y(t)\| \leq \tilde{\delta}.$$

Combining this with the distance between the curves and triangle inequality, we get

$$\|\tilde{y}(t) - y(\bar{t})\| \leq 2\tilde{\delta}.$$

Therefore, we showed that every point on the curve $\tilde{y}$ in time interval $[t_0, t_0 + h]$ is close to a point in the cover $\mathcal{C}$. Combining this with Equation (33) implies with probability at least $1 - 2\varepsilon_1$

$$\mathbb{P}\left( \sup_{t \in [t_0, t_0+h]} \left| \partial_t^k g_i(t) \right|_{t=0} \geq C_1 \right) \leq 2\varepsilon_1$$

$$\square$$

# F    PROOF OF PROPOSITION 11

In this section we exploit the framework we developed for obtaining a low-degree approximation of the score function on the probability flow ODE using the Picard iteration.

## F.1    PRELIMINARIES

Here we recall the basic setup for Picard iteration with polynomial approximation, as developed in Lee et al. (2018).

**Definition 26.** *Given vector field $F(x) : \mathbb{R}^d \to \mathbb{R}^d$, motivated by the Picard iteration and following Lee et al. (2018), define the operator $T(x)$ that acts on a curve $x(t) : [t_0, t_0 + h] \to \mathbb{R}^d$ as*

$$T(x)(t) = x(t_0) + \int_{t_0}^{t_0+h} F(x(s))ds.$$

*Furthermore, suppose we have a basis $\{\phi_j\}_{j=1}^D$ of smooth one dimensional functions and points $\{c_j\}_{j=1}^D$ such that $1 \le \forall i, j \le D$, $\phi_j(c_i) = 0$ if $i \ne j$ and $\phi_i(c_i) = 1$. Then, define the approximation $T_\phi$ of the $T$ operator corresponding to the basis $\{\phi_j\}_{j=1}^D$ by*

$$T_\phi(x)(t) = \int_{t_0}^{t_0+h} \sum_{j=1}^D F(x(c_j))\phi_j(s)ds. \tag{34}$$

In particular, we pick $\phi_j$'s to be a basis for one dimensional polynomials of degree less than $D$; we choose them as the Lagrange multiplier polynomials for points $c_j$. This way, the integral in (34) can be computed in closed form.

**Definition 27.** *We say the basis $\phi = \{\phi_j\}_{j=1}^D$ is $\gamma_\phi$ bounded if*

$$\sum_j \Big| \int_{t_0}^{t_0+h} \phi_j(s)\, ds \Big| \le \gamma_\phi$$

We further assume that the our estimate $F_t(x)$ for the score function is $\tilde{L}$ Lipschitz:

**Assumption 28.** *For all $x, y \in \mathbb{R}^d$, we have*

$$\|F_t(x) - F_t(y)\| \le \tilde{L} \|x - y\|.$$

Next, we generalize the approach of Lee et al. (2018) and show an important Lipschitz property of $T_\phi$ with respect to the supremum norm $\|.\|$, when the vector field $F$ is close to a Lipschitz one.

**Lemma 29.** *Suppose the basis $\phi$ is $\gamma_\phi$ bounded, and the vector field $F_s$ is $\tilde{L}$ Lipschitz. Then, for arbitrary $x, y \in \mathcal{C}([t_0, t_0 + h], \mathbb{R}^d)$ with $x(t_0) = y(t_0)$,*

$$\|T_\phi(x) - T_\phi(y)\|_{[t_0,t_0+h]} \le \tilde{L}\|x - y\|_{[t_0,t_0+h]}\gamma_\phi h,$$

$$\|T_\phi^{\circ\ell}(x) - T_\phi^{\circ\ell}(y)\|_{[t_0,t_0+h]} \le \Big(\tilde{L}\gamma_\phi h\Big)^\ell \|x - y\|_{[t_0,t_0+h]}. \tag{35}$$

*Proof.* For every $1 \le s \le h$, using $\tilde{L}$ Lipschitz property of our estimate for the score function,

$$\|F_t(x(t)) - F_t(y(t))\| \le \tilde{L}\|x(t) - y(t)\|$$
$$\le \tilde{L}\|x - y\|_{[t_0,t_0+h]}.$$

Hence

$$\|T_\phi(x) - T_\phi(y)\|_{[t_0,t_0+h]} = \sup_{t_0 \le h' \le t_0+h} \Big\| \int_{t_0}^{t_0+h'} \sum_{j=1}^D \Big( F_{c_j}(x(c_j)) - F_{c_j}(y(c_j)) \Big)\phi_j(s)ds \Big\|$$

$$\le \sup_{1 \le j \le D} \Big\| F_{c_j}(x(c_j)) - F_{c_j}(y(c_j)) \Big\|_\infty \sum_j \Big| \int_{t_0}^{t_0+h} \phi_j(s)ds \Big|$$

$$\le \tilde{L}\|x - y\|_{[t_0,t_0+h]}\gamma_\phi h.$$

the second line in (35) follows from applying the first line in (35) $\ell$ times. $\qquad\square$

**Definition 30.** *(Low-degree vector field) Let $y_t$ be the solution to the ODE $\dot{y}_t = F_t^*(y_t)$. We say the vector field $F^*$ is low-degree along $y_t$ if it accepts the following low degree approximation:*

$$\|F^* \circ y - P_{\leq D}(F^* \circ y)\|_{[t_0, t_0 + h]} \leq \varepsilon_{ld},$$

*where the curve $P_{\leq D}(F^* \circ y)$ is an approximation of $F^* \circ y$ whose coordinates are degree at most $D$ polynomials in the time variable $t \in [t_0, t_0 + h]$.*

**Lemma 31.** *For $\dot{y}_t = F_t^*(y_t)$, given that $F^*$ is a low-degree vector field based on Definition 30, we have*

$$\left\|T_\phi^{\circ m} y - y\right\| \leq \frac{\left(\tilde{L}\gamma_\phi h\right)^m - 1}{\left(\tilde{L}\gamma_\phi h\right) - 1} (\varepsilon_{ld} + \max_{j=1}^D \|F(y(c_j)) - F^*(y(c_j))\|)\,(1 + \gamma_\phi)\,h.$$

*Proof.* We can write $T_\phi(y) = y_{t_0} + S_\phi(F \circ y)$, where for arbitrary curve $z \in \mathcal{C}([t_0, t_0 + h], \mathbb{R}^d)$ define

$$S_\phi(z)(.) = \sum_{j=1}^D z(c_j) \int_{t_0}^{\cdot} \phi_j(s) ds.$$

Note that because $P_{\leq D}(F^* \circ y)$ is degree at most $D$ and $\phi_j$'s are the Lagrange multiplier polynomials at the Chebyshev points $(c_i)_{j=1}^D$, then

$$P_{\leq D}(F^* \circ y) = \sum_{j=1}^D P_{\leq D}(F^* \circ y)(c_j)\phi_j,$$

which from the definition of $S_\phi(z)$, implies

$$\int_{t_0}^{\cdot} P_{\leq D}(F^* \circ y)(s) ds = S_\phi(P_{\leq D}(F^* \circ y)).$$

Now combining this with Lemma 29 and using the low-degree Definition 30

$$\left\|S_\phi(F \circ y) - \int_{t_0}^{\cdot} P_{\leq D}(F^* \circ y)(s) ds\right\|_{[t_0, t_0 + h]} \tag{36}$$

$$= \|S_\phi(F \circ y) - S_\phi(P_{\leq D}(F^* \circ y))\|_{[t_0, t_0 + h]} \tag{37}$$

$$= \left\|\sum_{j=1}^D \left(F \circ y\big|_{c_j} - P_{\leq D}(F^* \circ y)\big|_{c_j}\right)\left(y_{t_0} + \int_{t_0}^{\cdot} \phi_j(s) ds\right)\right\|_{[t_0, t_0 + h]} \tag{38}$$

$$\leq (\max_{j=1}^D \|F^*(y(c_j)) - P_{\leq D}(F^*(y(c_j)))\| + \|F(y(c_j)) - F^*(y(c_j))\|) \sum_{j=1}^D \left\|y_{t_0} + \int_{t_0}^{\cdot} \phi_j(s) ds\right\|_{[t_0, t_0 + h]} \tag{39}$$

$$\leq (\varepsilon_{ld} + \max_{j=1}^D \|F(y(c_j)) - F^*(y(c_j))\|) \sum_{j=1}^D \left\|y_{t_0} + \int_{t_0}^{\cdot} \phi_j(s) ds\right\|_{[t_0, t_0 + h]} \tag{40}$$

$$\leq (\varepsilon_{ld} + \max_{j=1}^D \|F(y(c_j)) - F^*(y(c_j))\|)\gamma_\phi h. \tag{41}$$

But from the definition of $\varepsilon_{ld}$,

$$\|P_{\leq D}(F^* \circ y) - F^* \circ y\|_{[t_0, t_0 + h]} \leq \varepsilon_{ld}, \tag{42}$$

which using the identity $y - y_{t_0} = \int_{t_0}^{\cdot} F \circ y(s) ds$ implies

$$\left\|\int_{t_0}^{\cdot} P_{\leq D}(F^* \circ y)(s) ds - (y - y_{t_0})\right\|_{[t_0, t_0 + h]} = \left\|\int_{t_0}^{\cdot} P_{\leq D}(F^* \circ y)(s) ds - \int_{t_0}^{\cdot} F^* \circ y(s) ds\right\|_{[t_0, t_0 + h]} \tag{43}$$

$$\leq \varepsilon_{ld} h \tag{44}$$

Combining (41) and (44)

$$\|T_\phi(y) - y\|_{[t_0, t_0 + h]} \le \left\| T_\phi(y) - y_{y_{t_0}} - \int_{t_0}^{\cdot} P_{\le D}\left(F^* \circ y\right) ds \right\|_{[t_0, t_0 + h]}$$

$$+ \left\| \int_{t_0}^{\cdot} P_{\le D}\left(F^* \circ y\right) ds - (y - y_{t_0}) \right\|_{[t_0, t_0 + h]}$$

$$\le (\varepsilon_{ld} + \max_{j=1}^{D} \|F(y(c_j)) - F^*(y(c_j))\|) (1 + \gamma_\phi) h.$$

Now applying Lemma 29 $i - 1$ times

$$\left\| T_\phi^{\circ i}(y) - T_\phi^{\circ(i-1)}(y) \right\| \le \left(\tilde{L}\gamma_\phi h\right)^{i-1} \|T_\phi(y) - y\|_{[0,h]} \tag{45}$$

$$\le \left(\tilde{L}\gamma_\phi h\right)^{i-1} (\varepsilon_{ld} + \max_{j=1}^{D} \|F(y(c_j)) - F^*(y(c_j))\|) (1 + \gamma_\phi) h \tag{46}$$

Summing (46) for $i = 1, \ldots, m$:

$$\left\| T_\phi^{\circ m}(y) - y \right\| \le \sum_{i=1}^{m} \left\| T_\phi^{\circ i} y - T_\phi^{\circ(i-1)} y \right\|$$

$$\le \frac{\left(\tilde{L}\gamma_\phi h\right)^m - 1}{\left(\tilde{L}\gamma_\phi h\right) - 1} (\varepsilon_{ld} + \max_{j=1}^{D} \|F(y(c_j)) - F^*(y(c_j))\|) (1 + \gamma_\phi) h.$$

$\square$

**Corollary 32** (Effect of approximate Picard iterations). *For the exact probability flow ODE $\dot{y}_t = F_t^*(y_t)$, given that $F^*$ is a low-degree vector field based on Definition 30 and given $\tilde{L}\gamma_\phi h \le \frac{1}{2}$, for arbitrary curve $x \in \mathcal{C}([t_0, t_0 + h], \mathbb{R}^d)$ we have*

$$\left\| T_\phi^{\circ m}(y) - y \right\|_{[t_0, t_0 + h]} \le 2(\varepsilon_{ld} + \max_{j=1}^{D} \|F(y(c_j)) - F^*(y(c_j))\|) (1 + \gamma_\phi) h$$

$$\left\| T_\phi^{\circ m}(x) - T_\phi^{\circ m}(y) \right\|_{[t_0, t_0 + h]} \le \frac{1}{2^m} \|x - y\|_{[t_0, t_0 + h]}.$$

*Proof.* Directly from Lemmas 29 and 31. $\square$

**Lemma 33.** *Recall the definition of $F$ and $F^*$ from Section B. For initial point $\|\bar{y}_{t_0} - y_{t_0}\| \le \varepsilon_p$, define $y_s$, $\tilde{y}_s$, and $\bar{y}_s$ for $t_0 \le s \le t_0 + h$ as*

$$\dot{y}_s = F_s^*(y_s),$$
$$\dot{\tilde{y}}_s = F_s^*(\tilde{y}_s),$$
$$\bar{y}(s) = \bar{y}_{t_0} + (s - t_0)F_{t_0}(\bar{y}_{t_0}).$$

*Then, under the low degree Assumption 30 for the curve $\tilde{y}$ in the time interval $[t_0, t_0 + h]$, and picking step size $h = O(\frac{1}{1+R^2})$ and assuming $\tilde{L} \le \frac{1}{2}$ and $\sigma_t \ge 1$,*

$$\left\| y - T_\phi^{\circ m}(\bar{y}) \right\|_{[t_0, t_0 + h]} \le 2\varepsilon_p(1 + 2(1 + \gamma_\phi)h) + (\varepsilon_{ld} + \max_{j=1}^{D} \|F(y(c_j)) - F^*(y(c_j))\|) (1 + \gamma_\phi) h$$

$$+ \frac{h}{2^m} \left( \tilde{L}\varepsilon_p + \varepsilon_{err} + \sqrt{d}R(R^2 + \ln(1/\varepsilon_1)) + (2 + 4R^2)\varepsilon_p \right).$$

*Proof.* First, since we have the assumption $T - t_0 - h \ge 1$, we have $L_s \le (2 + 4R^2), \forall s \in [t_0, t_0 + h]$. Hence, since we know $\gamma_\phi = O(1)$ for the Chebyshev basis, the assumption $h \le O(\frac{1}{1+R^2})$ satisfies the precondition of Corollary 11 on $h$. On the other hand, from Lemma 37, again from the condition

$h = O(\frac{1}{1+R^2})$, we get $\|y - \tilde{y}\|_{[t_0, t_0+h]} \le 2\varepsilon_p$. Now combining this with Corollary 11:

$$\left\|y - T_\phi^{\circ m}(\bar{y})\right\|_{[t_0, t_0+h]} \le \|y - \tilde{y}\|_{[t_0, t_0+h]} + \left\|\tilde{y} - T_\phi^{\circ m}(\tilde{y})\right\|_{[t_0, t_0+h]} + \left\|T_\phi^{\circ m}(\bar{y}) - T_\phi^{\circ m}(\tilde{y})\right\|_{[t_0, t_0+h]}$$

$$\le 2\varepsilon_p + 2(\varepsilon_{ld} + \max_{j=1}^{D}\|F(\tilde{y}(c_j)) - F^*(\tilde{y}(c_j))\|)\,(1 + \gamma_\phi)\,h + \frac{1}{2^m}\|\bar{y} - \tilde{y}\|_{[t_0, t_0+h]}$$

$$\le 2\varepsilon_p + 2(\varepsilon_{ld} + 2\|y - \tilde{y}\|_{[t_0, t_0+h]} + \max_{j=1}^{D}\|F(y(c_j)) - F^*(y(c_j))\|)\,(1 + \gamma_\phi)\,h$$

$$+ \frac{1}{2^m}\|\bar{y} - \tilde{y}\|_{[t_0, t_0+h]}$$

$$\le 2\varepsilon_p(1 + 2(1 + \gamma_\phi)h) + 2(\varepsilon_{ld} + \max_{j=1}^{D}\|F(y(c_j)) - F^*(y(c_j))\|)\,(1 + \gamma_\phi)\,h$$

$$+ \frac{1}{2^m}\|\bar{y} - \tilde{y}\|_{[t_0, t_0+h]} \tag{47}$$

But from Lemma 38, with probability at least $1 - \varepsilon_1$, for all $t_0 \le s \le t_0 + h$

$$\|F_s^*(\tilde{y}_s)\| \le 6\left(R + \sqrt{d} + \sqrt{\ln(1/\varepsilon_1)}\right). \tag{48}$$

Hence, using Lemma 36 and 37,

$$\frac{d}{ds}\|\bar{y}_s - \tilde{y}_s\| \le \frac{\langle F_{t_0}(\tilde{y}_{t_0}) - F_s^*(\tilde{y}_s), \bar{y}_t - y_t\rangle}{\|\bar{y}_t - y_t\|}$$

$$\le \|F_{t_0}(\tilde{y}_{t_0}) - F_t^*(\tilde{y}_t)\|$$

$$\le \|F_{t_0}(\tilde{y}_{t_0}) - F_{t_0}(y_{t_0})\| + \left\|F_{t_0}(y_{t_0}) - F_{t_0}^*(y_{t_0})\right\|$$

$$+ \left\|F_{t_0}^*(y_{t_0}) - F_t^*(y_t)\right\| + \|F_t^*(y_t) - F_t^*(\tilde{y}_t)\|$$

$$\le \tilde{L}\varepsilon_p + \varepsilon_{err} + \sqrt{d}R(R^2 + \ln(1/\varepsilon_1)) + (2 + 4R^2)e^{\frac{5R^2}{\sigma_t^2}h}\varepsilon_p.$$

which implies

$$\|\bar{y} - \tilde{y}\|_{[t_0, t_0+h]} \le h\left(\left\|F_{t_0}(\tilde{y}_{t_0}) - F_{t_0}^*(\tilde{y}_{t_0})\right\| + 12\left(R + \sqrt{d} + \sqrt{\ln(1/\varepsilon_1)}\right)\right).$$

Plugging this back into (47)

$$\left\|y - T_\phi^{\circ m}(\bar{y})\right\| \le 2\varepsilon_p(1 + 2(1 + \gamma_\phi)h) + (\varepsilon_{ld} + \max_{j=1}^{D}\|F(y(c_j)) - F^*(y(c_j))\|)\,(1 + \gamma_\phi)\,h$$

$$+ \frac{h}{2^m}\left(\tilde{L}\varepsilon_p + \varepsilon_{err} + \sqrt{d}R(R^2 + \ln(1/\varepsilon_1)) + (2 + 4R^2)e^{\frac{5R^2}{\sigma_t^2}h}\varepsilon_p\right).$$

Using the assumption $h = O(\frac{1}{1+R^2})$ completes the proof. $\qquad\qquad\square$

# G   PROOF OF THEOREM 12

In this section, we put everything together to show our main end-to-end result, namely how to sample from the target distribution with TV error $\tilde{O}(\varepsilon_{err}R^2)$ and Wasserstein error $\tilde{O}(\varepsilon_{err})$.

**Theorem 34** (Formal version of Theorem 12). *For $\varepsilon_1$ and $\varepsilon_{err}$ satisfying $\varepsilon_{err} \le \varepsilon_1/\ln(d/R)$ and $\varepsilon_{err} \le \left((R/\sigma) \wedge (1/\sqrt{d})\right)/(\ln(\sigma/R) + \ln(d))^2$, given step size $h = \frac{\gamma}{k(1+(R/\sigma)^2)(\ln(1/\varepsilon_1)+\ln(d))}$ for small enough constant $\gamma$, with probability at least*

$$1 - O\left(\varepsilon_1 T(R/\sigma)^2 \ln(1/\varepsilon_{err})(\ln(1/\varepsilon) + \ln(d))\right),$$

*for $T = \ln(R/\sigma) + \ln(d) + \ln(1/\varepsilon_{err})$, Algorithm 2 outputs a sample whose distribution is close in 2-Wasserstein distance to the target measure within error of*

$$O(Tk\varepsilon_{err}\log(1/\varepsilon_1)) = O\left((\ln(R/\sigma) + \ln(d) + \ln(1/\varepsilon_{err}))\ln(1/\varepsilon_{err})\ln(k/\varepsilon_1)\varepsilon_{err}\right),$$

*in*

$$O\left(T(R/\sigma)^2 \ln(1/\varepsilon_{err})(\ln(1/\varepsilon_1) + \ln(d))\right)$$

*number of rounds and $m = \ln(R/\sigma) + \ln(d) + \ln\ln(1/\varepsilon_1) + \ln(1/\varepsilon_{err})$ number of Picard iterations in each round.*

*Proof.* We can scale the distribution, sample from the scaled distribution, then scale back. Note that this scaling procedure does not change the ratio $R/\sigma$. Therefore, without loss of generality, we assume $\sigma_0 = \sigma = 1$, since our bounds only depend on the quantity $R/\sigma$. We run the forward process up to time

$$T := \Theta(\ln(R) + \ln(d) + \ln(1/\varepsilon_{err})). \tag{49}$$

Now we prove inductively that $\left\| y - T_\phi^{\circ m}(\bar{y}) \right\|_{[t^{(i)}, t^{(i)}+h]} = O((t^{(i)} + h)k\varepsilon_{err} \log(k/\varepsilon_1))$ for all $i$.

First, note that we can bound the Wasserstein distance of the target distribution $q'$ and $\mathcal{N}(0, I)$ as

$$W_2(q', N(0, I)) \leq W_2(q', \delta_{\{0\}}) + W_2(\delta_{\{0\}}, N(0, I)),$$

where $\delta_{\{0\}}$ is the point mass at the origin. But

$$W_2(q', \delta_{\{0\}}) = \mathbb{E}_{Y \sim p_0} \|Y\|^2 \leq R,$$
$$W_2(N(0, I), \delta_{\{0\}}) = \sqrt{d}.$$

Therefore

$$W_2(q', N(0, I)) \leq R + \sqrt{d}.$$

Now from the Wasserstein contraction property of the OU process and the choice of $T$ in (49), we get

$$W_2(q_T, N(0, I)) \leq \left(R + \sqrt{d}\right) e^{-T} = O(\varepsilon_{err}).$$

To prove the step of induction for the interval $[t^{(i)}, t^{(i)} + h] = [t_0, t_0 + h]$, we know from the hypothesis of induction that for all previous intervals $(t^{(j)}, t^{(j)} + h)$ for $j < i$:

$$\left\| y - T_\phi^{\circ m}(\bar{y}) \right\|_{[t^{(j)}, t^{(j)}+h]} \leq O((t^{(j)} + h)k\varepsilon_{err} \log(k/\varepsilon_1)),$$

which from the definition of the iterates of the algorithm, i.e. $\hat{y} = T_\phi^{\circ m}(\bar{y})$ implies

$$\|y - \hat{y}\|_{[0,t_0]} \leq O(t_0 k\varepsilon_{err} \log(k/\varepsilon_1)).$$

Using Lemma 37, this further implies, given the choice of $h$,

$$\|\tilde{y} - y\|_{[t_0, t_0+h]} = O(t_0 k\varepsilon_{err} \log(k/\varepsilon_1)).$$

Hence, we can use $\varepsilon_p = O(t_0 k\varepsilon_{err} \log(k/\varepsilon_1))$ in Lemma 33.

Now given the assumptions $\varepsilon_{err} \leq (1/\ln(d/R))\varepsilon_1$ and $\varepsilon_{err} \leq \left(R \wedge (1/\sqrt{d})\right) / (\ln(1/R) + \ln(d))^2$, and since $\sigma_t \geq \sigma_0 \geq 1$ it is easy to check that $\tilde{\delta} = O(t_0 k\varepsilon_{err} \log(k/\varepsilon_1))$ satisfies the assumption of Lemma 25.

Hence, we can apply Lemma 25 with

$$k = \ln(1/\varepsilon_{err}),$$

using the fact that $\sigma_t \geq \frac{1}{2}$ and $R \geq 1$ there is constant $c_2$ such that for the interval $[t_0, t_0 + h] := [t^{(i)}, t^{(i)} + h]$,

$$\mathbb{P}\left(\sup_{s \in [t_0, t_0+h]} \left| \partial_s^k g_i^{(\tilde{y}(s))}(s) \right|_{s=0} \geq \left(c_2 k^2 R^2 \left(\ln(1/\varepsilon_1) + \ln(d)\right)\right)^k\right) \leq 2\varepsilon_1.$$

Moreover, using this with Fact C.1 and by picking step size

$$h = \frac{\gamma}{k(1 + R^2)\left(\ln(1/\varepsilon_1) + \ln(d)\right)}$$

for small enough constant $\gamma$, we get with probability at least $1 - 2\varepsilon_1$

$$\|g - P_{\leq k}g\|_{[t_0, t_0+h]} \leq \left(hc_2 e k R^2 \left(\ln(1/\varepsilon_1) + \ln(d)\right)\right)^k \leq \varepsilon_{err}. \tag{50}$$

Therefore, we can now use Lemma 33 with

$$m := \ln(R) + \ln(d) + \ln\ln(1/\varepsilon_1) + \ln(1/\varepsilon_{err}) + \ln(\tilde{L})$$

number of Picard iterations; using the fact that $\gamma_\phi = O(1)$ for the Chebyshev basis, setting $\varepsilon_{ld} = \varepsilon_{err}$ in Assumption 30 based on Equation (50), and using Assumption 3, we get

$$\left\| y - T_\phi^{\circ m}(\bar{y}) \right\|_{[t_0, t_0+h]} = O\left(\varepsilon_p + h\left(\varepsilon_{err} + \sum_{j=1}^{k} \|F(y(c_j)) - F^*(y(c_j))\|\right)\right) \tag{51}$$

$$+ \frac{h}{2^m} O\left(\tilde{L}\varepsilon_p + \varepsilon_{err} + \sqrt{d}R(R^2 + \ln(1/\varepsilon_1)) + (2 + 4R^2)\varepsilon_p\right) \tag{52}$$

$$\leq O(\varepsilon_p + hk\varepsilon_{err}\log(k/\varepsilon_1)) \tag{53}$$

$$+ \frac{h}{2^m} O\left(\tilde{L}\varepsilon_p + \varepsilon_{err} + \sqrt{d}R(R^2 + \ln(1/\varepsilon_1)) + (2 + 4R^2)\varepsilon_p\right)., \tag{54}$$

where the last line follows from the choice of $m$.

Now using the fact that $h = O(\frac{1}{1+R^2})$, we have

$$\frac{h}{2^m}(2 + 4R^2)\varepsilon_p = O(\frac{1}{2^m}Tk\varepsilon_{err}\log(1/\varepsilon_1)) = O(\frac{T/h}{2^m}hk\varepsilon_{err}\log(1/\varepsilon_1)) = O(hk\varepsilon_{err}\log(k/\varepsilon_1)),$$

where we used the fact that

$$T/h = O\left((1 + R^2)\ln(1/\varepsilon_{err})^2(\ln(1/\varepsilon_1) + \ln(d))\right) = O(2^m).$$

Similarly, it is not hard to check that with the choice of $m$ we have

$$\frac{h}{2^m}\left(\tilde{L}\varepsilon_p + \varepsilon_{err} + \sqrt{d}R(R^2 + \ln(1/\varepsilon_1))\right) = O(hk\varepsilon_{err}\log(k/\varepsilon_1))/$$

Therefore, we can upper bound (54) as

$$\mathbb{E}\left\| y - T_\phi^{\circ m}(\bar{y}) \right\|_{[t_0, t_0+h]} \leq O(\varepsilon_p) + O(hk\varepsilon_{err}\log(k/\varepsilon_1)) = O((t_0 + h)k\varepsilon_{err}\log(k/\varepsilon_1)).$$

which proves the step of induction. Therefore, overall we showed

$$\|y - \hat{y}\|_{[0,T]} = O(Tk\varepsilon_{err}\log(1/\varepsilon_1)) = O\left((\ln(R) + \ln(d) + \ln(1/\varepsilon_{err}))\ln(1/\varepsilon_{err})\log(k/\varepsilon_1)\varepsilon_{err}\right),$$

after $O\left(TR^2\ln(1/\varepsilon_{err})(\ln(1/\varepsilon_1) + \ln(d))\right)$ number of rounds, where in each round we apply the Picard iteration for $m = \ln(R) + \ln(d) + \ln\ln(1/\varepsilon_1) + \ln(1/\varepsilon_{err})$ number of times. Note that this Wasserstein guarantee holds only with probability at least $1 - O(\varepsilon_1 T/h) = 1 - O\left(\varepsilon_1 TR^2\ln(1/\varepsilon_{err})(\ln(1/\varepsilon) + \ln(d))\right)$ after applying a union bound. $\square$

## H PROOF OF COROLLARY 14

In this section we show how to use underdamped Langevin Monte Carlo to upgrade the Wasserstein bound achieved by our sampler into a total variation bound, thus proving Corollary 14. The pseudocode for this is provided in Algorithm 3, where we propose a slight modification of Algorithm 2 that relies on running the underdamped Langevin Monte Carlo algorithm (see Appendix H.1) for a short period of time at the end of Algorithm 2. This step is often referred to as a *corrector step* in the diffusion model literature. The resulting Algorithm 3 admits a TV guarantee, stated in Corollary 14.

### H.1 UNDERDAMPED LANGEVIN MONTE CARLO

In this section, for the sake of completeness, we briefly review underdamped Langevin Monte Carlo, which is only used in this final phase of our algorithm to convert from Wasserstein closeness to TV closeness.

---

**Algorithm 3:** CORRECTEDCOLLOCATIONDIFFUSION$((s_t), \varepsilon)$

---

**Input:** Score estimates $s_t$ satisfying Assumption 3, target error $0 < \varepsilon < 1$

**Output:** Sample from a distribution $\hat{q}$ satisfying $\mathrm{TV}(\hat{q}, q) \leq \varepsilon$

1 $\eta \leftarrow \varepsilon^{5/3}/(L^{1/4}d^{1/2})$, where $L \leq \mathrm{poly}(d/\eta)$ is an upper bound on the Lipschitz constant of $q$.

2 $x \leftarrow$ COLLOCATIONDIFFUSION$((s_t), \varepsilon)$

3 Run underdamped Langevin Monte Carlo (see Appendix H.1) with friction parameter $\Theta(\sqrt{L})$
  and step size $h = \varepsilon^{2/3}/(d^{1/3}M(\eta)^{1/3}L^{1/2})$ for $M(\eta)$ steps, where $M(\eta)$ denotes the number
  of steps used to run COLLOCATIONDIFFUSION in the previous step. Let the resulting sample be
  $x'$.

4 **return** $x'$

---

Given an estimate $s$ of the log-density of a distribution $q$, and a *friction parameter* $\gamma$, underdamped Langevin Monte Carlo with step size $h$ and score estimate $s$ is given by

$$\mathrm{d}x_t = v_t\,\mathrm{d}t$$
$$\mathrm{d}v_t = (s(x_{\lfloor t/h \rfloor h}) - \gamma v_t)\,\mathrm{d}t + \sqrt{2\gamma}\,\mathrm{d}B_t\,,$$

where $B_t$ is a standard Brownian motion.

We will use the following result of Chen et al. (2023b), which is a consequence of the short-time regularization of Guillin & Wang (2012):

**Theorem 35** (Theorem A.5 of Chen et al. (2024c), restated)**.** *Let $q \propto e^{-H}$ be a distribution over $\mathbb{R}^d$ for which $\nabla H$ is $L$-Lipschitz. Let $p$ be an arbitrary distribution over $\mathbb{R}^d$. Suppose $s : \mathbb{R}^d \to \mathbb{R}^d$ satisfies $\|s - \nabla H\|_{L_2(q)}^2 \leq \varepsilon_{\mathrm{sc}}^2$. Let $T \lesssim 1/\sqrt{L}$.*

*If $p_N$ denotes the distribution given by running underdamped Langevin Monte Carlo initialized at $p$ for $T/h$ steps and step size $h$ with friction parameter $\Theta(\sqrt{L})$, then*

$$\mathrm{TV}(p_N, q) \lesssim \frac{W_2(p, q)}{L^{1/4}T^{3/2}} + \frac{\varepsilon_{\mathrm{sc}}T^{1/2}}{L^{1/4}} + L^{3/4}T^{1/2}d^{1/2}h\,. \tag{55}$$

### H.2 PROOF OF COROLLARY 14

*A priori* it might appear that the third term in Eq. (55) forces a choice of $h = O(d^{-1/2}\varepsilon)$, translating to an iteration complexity of $\Omega(d^{1/2}/\varepsilon)$. Here we use an idea of Gupta et al. (2024): because in our application of Theorem 35, $W_2(p, q)$ can be made quite small, we can take $T$ to be small to get a much better bound on the iteration complexity $T/h$.

*Proof of Corollary 14.* For $\eta$ to be tuned, let $M(\eta) = \mathrm{poly}(R/\sigma) \cdot \log(1/\eta)$ denote the number of iterations of Algorithm 2 needed to achieve Wasserstein error $\eta$ in Theorem 12. We will take $h = \varepsilon^{2/3}/(d^{1/3}M(\eta)^{1/3}L^{1/2})$ and $T = M(\eta)h$ in Theorem 35 to conclude that, starting from the distribution given by Algorithm 2, if we run underdamped Langevin Monte Carlo for $M(\eta)$ iterations, we will produce a distribution $p_N$ for which

$$\mathrm{TV}(p_N, q) \lesssim \frac{\eta L^{1/4}d^{1/2}}{M(\eta)^{2/3}\varepsilon^{2/3}} + \frac{\varepsilon_{\mathrm{sc}}M(\eta)^{1/3}\varepsilon^{1/3}}{L^{1/2}d^{1/6}} + \varepsilon\,. \tag{56}$$

If we take $\eta = \varepsilon^{5/3}/(L^{1/4}d^{1/2})$, then $M(\eta) \leq (R/\sigma)^C \log(Ld/\varepsilon)$ and the first term on the right-hand side is bounded by $\varepsilon$. By the assumption that $\varepsilon_{\mathrm{sc}} \leq \tilde{O}(\frac{\varepsilon^{2/3}L^{1/2}d^{1/6}}{\mathrm{poly}(R/\sigma)^{1/3}})$, the second term on the right-hand side is bounded by $\varepsilon$. By replacing $\varepsilon$ with $c\varepsilon$ for sufficiently small constant $c$ in the above, we obtain the claimed bound. $\qquad\square$

## I OTHER USEFUL ESTIMATES

### I.1 UPPER ESTIMATES ON THE MOVEMENT OF THE PROBABILITY FLOW ODE

In this section we prove some useful properties of the score function. First, we bound the smoothness of the score function $\nabla \log q_t$.

**Lemma 36.** *(Bounding the operator norm of the true score) For $t \geq 1$, we have*

$$\|\nabla (y_t + \nabla \log q_t(y_t))\| \leq e^{-(T-t)} \left(2 + 4R^2\right),$$

$$\left\|\nabla^2 \log q_t(y_t)\right\|_{\mathsf{op}} \leq 2 + 4e^{-(T-t)} R^2,$$

*and for $2(T - t) < 1$,*

$$\|\nabla (y_t + \nabla \log q_t(y_t))\| \leq \frac{5R^2}{(T-t)^2}.$$

*Proof.* It is easy to check the Hessian of $\log q_t$ can be written in the following form:

$$-\nabla^2 \log q_t(y) = -\frac{1}{\sigma_t^4} \mathrm{Cov}(q^{t,y}) + \frac{1}{\sigma_t^2} I.$$

But using the radius $R$ assumption on the support of $p_0$, we can bound the covariance as

$$\mathrm{Cov}(q^{t,y}) \leq \frac{1 - \sigma_t^2}{\sigma_t^4} R^2 I.$$

Therefore

$$-\nabla (y_t + \nabla \log q_t(y_t)) = \frac{1 - \sigma_t^2}{\sigma_t^2} I - \frac{1 - \sigma_t^2}{\sigma_t^4} R^2 I. \tag{57}$$

Now for $t \geq 1$, we have $\sigma_t^2 = (1 - e^{-2(T-t)}) \geq 1 - 1/e^2 \geq 0.5$. Therefore, for $t \geq 1$,

$$\|\nabla (y_t + \nabla \log q_t(y_t))\| \leq (1 - \sigma_t^2) \left(2 + 4R^2\right) = e^{-2(T-t)} \left(2 + 4R^2\right).$$

Similarly

$$\left\|\nabla^2 \log q_t(y_t)\right\| \leq 2 + (1 - \sigma_t^2) \left(4R^2\right) = 2 + 4e^{-2(T-t)} R^2.$$

On the other hand, for $s = 2(T - t) < 1$, using $e^{-s} \leq 1 - s + \frac{s^2}{2} \leq 1 - \frac{s}{2}$ we have $\sigma_t^2 = 1 - e^{-(T-t)} \geq T - t$. Then from the assumption $R \geq 1$ and Equation (57), we have

$$\|\nabla (y_t + \nabla \log q_t(y_t))\| \leq \left(\frac{2}{T-t} + \frac{4R^2}{(T-t)^2}\right) \leq \frac{5R^2}{(T-t)^2}. \qquad \square$$

We can then use this bound on the smoothness to control the extent to which two processes evolving according to the same probability flow ODE diverge over time:

**Lemma 37.** *(Distance between true ODE solutions starting from close points) For $y_t, \tilde{y}_t$ that evolve according to probability flow ODE, i.e.,*

$$\frac{d}{dt} y_t = y_t + \nabla \log q_t(y_t)$$

$$\frac{d}{dt} \tilde{y}_t = \tilde{y}_t + \nabla \log q_t(\tilde{y}_t),$$

*with initial condition satisfying $\|y_{t_0} - \tilde{y}_{t_0}\| \leq \varepsilon$, then, for time window $s$ such that $T - (t_0 + s) \geq 1$, we have*

$$\|y_{t_0+s} - \tilde{y}_{t_0+s}\| \leq e^{\frac{5R^2}{\sigma_t^2} s} \varepsilon$$

*Proof.* Note that for $t_0 \leq t \leq t_0 + s$, we have $t = T - t \geq 1$ from our assumption, therefore from Lemma 36:

$$\frac{d}{dt} \|y_t - \tilde{y}_t\| = \frac{\langle y_t - \tilde{y}_t, \nabla \log q_t(y_t) - \nabla \log q_t(\tilde{y}_t)\rangle}{\|y_t - \tilde{y}_t\|}$$

$$= \frac{\int_{r=0}^{1} (y_t - \tilde{y}_t)^\top \nabla (y_t + q_t(y_t + r(\tilde{y}_t - y_t))) (y_t - \tilde{y}_t) \, dr}{\|y_t - \tilde{y}_t\|}$$

$$\leq e^{-t}(2 + 4R^2) \frac{\|y_t - \tilde{y}_t\|^2}{\|y_t - \tilde{y}_t\|}$$

$$= e^{-t}(2 + 4R^2) \|y_t - \tilde{y}_t\|, \tag{58}$$

which implies

$$\frac{d}{dt}\ln\left(\|y_t - \tilde{y}_t\|\right) \le e^{-t}(2 + 4R^2). \tag{59}$$

Integrating Equation (59) from $t = t_0$ to $t = t_0 + s$ we get the desired result for the first part. Similarly for the case when $T - t_0 < 1$:

$$\frac{d}{dt}\ln\left(\|y_t - \tilde{y}_t\|\right) \le \frac{5R^2}{t^2}, \tag{60}$$

which implies (using inequality $\sigma_t^2 = 1 - e^{-t} \le t$ for $t = T - (t_0 + s)$)

$$\|y_{t_0+s} - \tilde{y}_{t_0+s}\| \le e^{\frac{5R^2}{T-(t_0+s)}s}\varepsilon \le e^{\frac{5R^2}{\sigma_t^2}s}\varepsilon. \qquad \square$$

**Lemma 38.** *(Distributional guarantees for probability flow ODE) Along the backward ODE*

$$\dot{y}_t = F_t^*(y_t) \tag{61}$$

*with $F_t^*(y) = y + \nabla \log q_t(y)$ we have with probability $1 - \varepsilon_1$,*

$$\forall t \in [t_0, t_0 + h], \|y_t\| \le e\left(R + \sqrt{d} + \sqrt{\ln(1/\varepsilon_1)}\right),$$

$$\forall t \in [t_0, t_0 + h], \|F_t^*(y_t)\| \le 6\left(R + \sqrt{d} + \sqrt{\ln(1/\varepsilon_1)}\right),$$

$$\forall s_1, s_2, \|y_{s_1} - y_{s_2}\| \le 6(s_2 - s_1)\left(R + \sqrt{d} + \sqrt{\ln(1/\varepsilon_1)}\right)$$

$$\left\|F_{t_0}^*(y_{t_0}) - F_{t_0+h}^*(y_{t_0+h})\right\| \lesssim \sqrt{d}R(R^2 + \ln(1/\varepsilon)).$$

*Proof.* Recall we can write $y_t = X_t + \sigma_t \xi$ where $F_t^*(y) = y_t + \frac{\mathbb{E}^{t,y}X - y_t}{\sigma_t^2}$. Now for time $t_0$, using $\sigma_{t_0}^2 \ge 0.5$ for $t_0 \ge 1$ as we showed in Lemma 36, we have

$$\left\|F_{t_0}^*(y_{t_0})\right\| = \mathbb{E}\left\|y_{t_0} + \frac{\mathbb{E}^{t,y_{t_0}}X - y_{t_0}}{\sigma_{t_0}^2}\right\| \tag{62}$$

$$\le \left(1 + \frac{1}{\sigma_{t_0}^2}\right)\|y_{t_0}\| + \frac{1}{\sigma_{t_0}^2}R \tag{63}$$

$$\le \frac{3}{2}\|y_{t_0}\| + \frac{R}{2} \tag{64}$$

Now because $\|\xi\|$ is subgaussian, we get with probability at least $1 - \varepsilon_1$,

$$\|y_{t_0}\| \le R + \sqrt{d} + \sqrt{\ln(1/\varepsilon_1)}.$$

On the other hand, from the definition (61)

$$\frac{d}{ds}\|y_s\| \le \|F_s^*(y_s)\| \le \frac{3}{2}\|y_s\| + \frac{R}{2},$$

so up to time $s$, we get

$$\|y_s\| \le (R + \sqrt{d} + \sqrt{\ln(1/\varepsilon_1)})e^{s\left(R+\sqrt{d}+\sqrt{\ln(1/\varepsilon_1)}\right)},$$

which implies from the fact that $h \le \frac{1}{R+\sqrt{d}+\sqrt{\ln(1/\varepsilon_1)}}$,

$$\|y_s\| \le e\left(R + \sqrt{d} + \sqrt{\ln(1/\varepsilon_1)}\right).$$

The second part follows from (64). For the third part we have

$$\|y_{s_1} - y_{s_2}\| = \left\|\int_{s_1}^{s_2} F_s^*(y_s)ds\right\|$$

$$\le \int_{s_1}^{s_2} \|F_s^*(y_s)\|\,ds$$

$$\le 6(s_2 - s_1)\left(R + \sqrt{d} + \sqrt{\ln(1/\varepsilon_1)}\right).$$

For the last part, using Lemma 8:

$$\left(\mathbb{E}\left\|\partial_t F_t^*(y_t)\right\|^p\right)^{1/p} \lesssim \sqrt{d}R(R + \sqrt{p})^2.$$

Integrating from $t_0$ to $t_0 + h$:

$$\left(\mathbb{E}\left\|F_{t_0}^*(y_{t_0}) - F_{t_0+h}^*(y_{t_0+h})\right\|^p\right)^{1/p} \lesssim h\sqrt{d}R(R + \sqrt{p})^2,$$

which implies with probability at least $\varepsilon_1$ we have

$$\left\|F_{t_0}^*(y_{t_0}) - F_{t_0+h}^*(y_{t_0+h})\right\| \lesssim \sqrt{d}R(R + \sqrt{\ln(1/\varepsilon)})^2.$$

This completes the proof. $\qquad\qquad\qquad\qquad\qquad\qquad\qquad\qquad\square$