# OpenReview forum: "High-accuracy and dimension-free sampling with diffusions"
_ICLR.cc/2026/Conference — Submitted to ICLR 2026_

### Official Review · Reviewer_r9Fo · 2025-10-26

**Soundness:** 3
**Presentation:** 3
**Contribution:** 3
**Rating:** 6
**Confidence:** 3

**Summary:**

This paper introduces a new diffusion-based sampling algorithm that achieves high accuracy (logarithmic dependence on error tolerance ($1/\epsilon$)).
The authors adapt the collocation method to diffusion models, showing that the score function’s time evolution along the probability flow ODE can be approximated by a low-degree polynomial. This enables long, stable integration steps without dimension-dependent error growth.
Under mild assumptions (bounded-support distribution convolved with Gaussian noise), they prove that their sampler reaches total variation distance $\epsilon$ from the target in
$(R/\sigma)^2 \log(1/\epsilon)$
iterations, where $R$ is the effective radius and $\sigma$ the noise level.
The authors claim that this is the first diffusion sampler with polylogarithmic dependence on ($1/\epsilon$) and dimension-free complexity. Theoretical guarantees are supported by rigorous convergence proofs and extensions to Wasserstein and TV metrics.

**Strengths:**

* **Theoretical novelty:** Provides the first high-accuracy, dimension-free complexity bound for diffusion sampling.
* **Elegant mathematical analysis:** Derives explicit bounds on higher-order time derivatives of the score, enabling low-degree polynomial approximation.
* **Clarity of assumptions:** The “bounded plus noise” assumption is realistic and aligns with practical diffusion model setups.
* **Practical relevance:** Insights into discretization stability and polynomial convergence are directly relevant to improving real-world diffusion solvers.

**Weaknesses:**

* **Strong assumptions:** Requires sub-exponential tails for score estimation error.
* **No empirical validation:** Results are purely theoretical; no experiments confirm practical speedups or quality gains.
* **Restricted generality:** The framework may not apply cleanly to unbounded or heavy-tailed data distributions common in real applications.
* **Method complexity:** Implementation of the collocation solver may be nontrivial compared to standard discretization schemes.

**Questions:**

- How sensitive is the convergence rate to violations of the sub-exponential error assumption in practice?
- Does the method maintain stability when applied to very high-dimensional or multimodal image data distributions?
- Could similar dimension-free guarantees hold under weaker moment conditions rather than bounded support?
- Are there empirical results planned to validate these theoretical claims on benchmark diffusion models?

---

> ### Author Response · Authors · 2025-11-19
> **Addressing potential weaknesses**
>
> Thanks for your review.
>
> **Strong assumptions: Requires sub-exponential tails for score estimation error.**
>
> This assumption can appear to be a weakness, however, as written in the footnote Page 4 (see below), we believe this assumption is necessary. Indeed, to reach high accuracy, some accuracy in the score estimation is needed.
>
> More precisely, if the tails were heavier, then there is an event, whose probability can be lower bounded, on which a fraction of the score estimates that we use in the algorithm are inaccurate by a constant amount. On this event, the sampler can deviate sufficiently that it will fail to converge in polylog(1/eps) many steps. We suspect that this is fundamental and leave proving a suitable lower bound as an open question.
>
>
>
> **Restricted generality: The framework may not apply cleanly to unbounded or heavy-tailed data distributions common in real applications.**
>
> Why this comment? As you wrote above (in the Strengths), our target (bounded + noise) is realistic. We can justify further if you clarify your question. Thanks in advance.
>
> **No empirical validation: Results are purely theoretical; no experiments confirm practical speedups or quality gains.**
>
> We prefer to keep our paper methodological to avoid additional complexity such as engineering tricks in numerical experiments etc. Our paper already conveys several complex and new ideas.
>
> **Method complexity: Implementation of the collocation solver may be nontrivial compared to standard discretization schemes.**
>
> We respectfully disagree. For example, our Algorithm 2 is only a few lines of code.
>
> That being said, we agree that standard discretization schemes are well understood in practice, engineering tricks are known etc. However, we believe this is a strength: we propose a new promising methodology that will be explored and engineered by the applied community and that can lead to important improvements.
>
> *We hope our reply answers your concerns. If that is the case, we would be grateful if you could raise your score.*

---

> > ### Author Response · Authors · 2025-11-19
> > **Answering questions**
> >
> > **How sensitive is the convergence rate to violations of the sub-exponential error assumption in practice?**
> >
> > Good question. As written above, we do not think one can obtain high accuracy (polylog) results without the sub-exponential error assumption. In fact, poly(1/eps) complexity has been obtained in the literature without sub-exponential error assumption.
> >
> > **Does the method maintain stability when applied to very high-dimensional or multimodal image data distributions?**
> >
> > Yes. Multimodal and high-dimensional are precisely the type of distributions we target
> >
> > 1. High-dimensional: Our complexity result does not depend explicitly on the dimension
> > 2. Multimodal. We target distributions which are bounded + noise (see Assumption 1). A typical case is a mixture of Gaussians, which corresponds to the bounded distribution being a sum of Diracs.
> >
> > **Could similar dimension-free guarantees hold under weaker moment conditions rather than bounded support?**
> >
> > We believe that in the most general case in our problem setting, some dependency on the diameter of the support is necessary (in the worst case). However, for further structured distributions like mixture of Gaussian we believe the dependency on the support diameter is not necessary by incorporating other subroutines in the algorithm to exploit that specialized structure.
> >
> > **Are there empirical results planned to validate these theoretical claims on benchmark diffusion models?**
> >
> > Yes, but empirical results are left for future work.

---

> > > ### Comment · Reviewer_r9Fo · 2025-11-27
> > > **Reply to authors**
> > >
> > > I thank the authors for their reply. I will maintain my current score.

---

### Official Review · Reviewer_Z2pA · 2025-10-28

**Soundness:** 3
**Presentation:** 2
**Contribution:** 3
**Rating:** 6
**Confidence:** 3

**Summary:**

This work introduces a new diffusion model solver based on the collocation method. The proposed solver achieves logarithmic scaling in inverse accuracy and avoids explicit dependence on dimensionality. It is the first high-accuracy diffusion-based sampler that operates with only approximate score information.

**Strengths:**

(1) The paper proposes a new solver based on the collocation method for diffusion models. A delicate design will lead to high-accuracy sampling.

(2) It proves that a polynomial iteration complexity on $1/\epsilon$, where $\epsilon$ is the sampling accuracy.

(3) The paper provides a detailed theoretical analysis.

**Weaknesses:**

(1) The introduction of collocation methods contains much ambiguity. For example, “ by polynomial interpolation”, in my understanding, polynomial interpolation first determines where it takes values ($c_i$), then we can find corresponding polynomials. The argument first finding a polynomial basis, then selecting nodes satisfying the equations, seems weird.

(2) Many definitions of norms are not specified in this paper. For example, Lemma 8, did I miss where the $||\cdot||_ {p,\infty}$ is defined? In lemma 9, $||y-\tilde{y} ||$, what is the norm here? Then in Theorem 10, the norm becomes $||\cdot ||_{\infty}$. Should it be the same as that in Lemma 8? I recommend that the readers write a section in the main text, including all the notations used.

(3) Theorem 10: the result mixes different $\delta$, one is the error of initialization distribution, another is the probability (at least $1-\delta$).

(4) The proof of the paper is very hard to read. The technical lemmas, for example, Lemma 6, look horrible. Could you provide a proof sketch of the main theorem? It should discuss how the technical lemmas are applied and why they are necessary.

(5) What is $\epsilon_{sc}$ in Cor 14? In Theorem 14, why you require $\epsilon_{err} \ge \epsilon$, instead of $<$?

(6) Line 1257, Lemma ??? is not exhibited.

(7) The paper misses some important related work, e.g., [1][2].

[1] Unified Convergence Analysis for Score-Based Diffusion Models with Deterministic Samplers ICLR Li et al.2024

[2] Convergence analysis of probability flow ode for score-based generative models Huang et al. 2024

**Questions:**

The contribution of this paper is more to apply the collocation method (a numerical method) to the ODE analysis of diffusion models, rather than proposing something specific to diffusion. Do you try to implement the algorithm in practice? How will it behave?

---

> ### Author Response · Authors · 2025-11-20
> **Addressing main potential weaknesses**
>
> Thanks for your review. We will reply in two times. First we address the main potential weaknesses of our work, then we will address items in the weakness section which are rather questions, or which can be addressed by clarifying the notations or adding references.
>
> **Main potential weaknesses**
>
> **(1) The introduction of collocation methods contains much ambiguity. For example, “ by polynomial interpolation”, in my understanding, polynomial interpolation first determines where it takes values $c_i$, then we can find corresponding polynomials. The argument first finding a polynomial basis, then selecting nodes satisfying the equations, seems weird.**
>
> Indeed, in practice one first select the nodes and then the polynomials. So, we will rewrite this sentence as follows: "Let $c_i \in [0,H]$, by polynomial interpolation there exists $\phi_j$ such that $\phi_j(c_i) = 1_{i=j}$"
>
> **(4) The proof of the paper is very hard to read. The technical lemmas, for example, Lemma 6, look horrible. Could you provide a proof sketch of the main theorem? It should discuss how the technical lemmas are applied and why they are necessary.**
>
> This is an important one, that we will clarify. Maybe we can resolve that by adding a title to each Lemma. The reader should focus on the meaning of the lemmas rather than the combinatorial upper bounds.
>
> To summarize, Lemma 6 expresses the first time derivative of the vector field along the trajectory. By iterating, we obtain Lemma 7 which expresses the k-th time derivative of the vector field along the trajectory. By upper bounding the r.h.s. in Lemma 7, we obtain Lemma 8 which bounds the k-th time derivative of the vector field along the trajectory. This bound means that the vector field along the trajectory has a small k-th derivative, therefore is well approximated by a polynomial. To derive our method, we can therefore approximate the vector field along the trajectory by a polynomial, and run collocation with this polynomial.
>
> **(5) What is $\varepsilon_{sc}$ in Cor 14? In Theorem 14, why you require $\varepsilon_{err} > \varepsilon$, instead of $<$?**
>
> $\varepsilon_{sc}$ is a typo, it should be $\varepsilon_{err}$.
>
> The value of $\varepsilon_{err}$ is given, the user does not control it, and it measures the accuracy of the approximation of the score by a neural network. Our paper proves high accuracy results, which means results that hold when $\varepsilon$ is small enough, in particular here $\varepsilon < \varepsilon_{err}$.

---

> > ### Author Response · Authors · 2025-11-24
> > **Addressing other points**
> >
> > We now address the other items.
> >
> > **(2) Many definitions of norms are not specified in this paper. For example, Lemma 8, did I miss where the** $||.||_{p,\infty}$ **is defined? In lemma 9, $||.||$, what is the norm here? Then in Theorem 10, the norm becomes $||.||_{\infty}$. Should it be the same as that in Lemma 8? I recommend that the readers write a section in the main text, including all the notations used.**
> >
> > Throughout the paper, $||.||$ denotes the Euclidean norm. The other norms are defined in the Notation paragraph (line 163).
> >
> > There is no inconsistency between Lemma 8 and Theorem 10. Lemma 8 is a statement in expectation involving an $L^p$ norm whereas Theorem 10 is a statement with high probability.
> >
> > **(3) Theorem 10: the result mixes different $\delta$, one is the error of initialization distribution, another is the probability (at least $1-\delta$).**
> >
> > Yes this is just a typo. Thanks for spotting.
> >
> > **(6) Line 1257, Lemma ??? is not exhibited.**
> >
> > This compilation error is because we submitted the Appendix without the main paper as supplementary material. Everything will be resolved by submitting a single file containing main paper and Appendix
> >
> > **(7) The paper misses some important related work, e.g., [1][2].**
> >
> > Thanks for these references. We will add them and discuss them in the literature review.
> >
> >
> > **The contribution of this paper is more to apply the collocation method (a numerical method) to the ODE analysis of diffusion models, rather than proposing something specific to diffusion. Do you try to implement the algorithm in practice? How will it behave?**
> >
> > The contribution of the paper is to show that a variant of collocation, *when applied to diffusion models,* provably achieves logarithmic complexity, without explicit dependence of the dimension. We believe this is a significant theoretical contribution. Experimental analysis is left for future work.
> >
> > *We hope to have answered your concerns. If that is the case, we would be grateful if you could raise your score.*

---

> ### Comment · Reviewer_Z2pA · 2025-11-25
>
> Thank you for your reply. The rebuttal has addressed most of my concerns. However, even with titles added to the lemmas, the theoretical lemma remains difficult to follow, particularly for readers from diverse backgrounds. I would recommend providing a more detailed and accessible description of the proof in the revision.
>
> In addition, the lack of empirical validation remains a notable weakness. The rebuttal does not sufficiently justify this point, as it does not discuss the practical challenges of implementing the algorithm. This raises concerns for me that the method may perform poorly in practice, which in turn casts doubt on its theoretical value.
>
> Overall, while the paper contributes valuable theoretical insights, the mentioned weaknesses persist. I will maintain my score and lean toward borderline acceptance.
>
> (As a minor note, some equations in the main text are overly long and extend beyond the page margins. This should be corrected in the revision.)
>
> Best,
> Reviewer Z2pA

---

> ### Author Response · Authors · 2025-11-26
> **Thanks for the reply**
>
> We appreciate that you took some time to reply to the rebuttal. This is the first reply we receive.
>
> **However, even with titles added to the lemmas, the theoretical lemma remains difficult to follow, particularly for readers from diverse backgrounds. I would recommend providing a more detailed and accessible description of the proof in the revision.**
>
> The proof technique is indeed difficult but there is not much we can do about it: that's the nature of the problem. The problem is difficult, and getting log complexity without dimension dependency was difficult.
>
> We provided an accessible description of the proof in our reply above (reproduced below), and we will add it to the paper if you request us to do so. We believe that you would judge anything more detailed than this description as too technical for a diverse audience.
>
> "To summarize, Lemma 6 expresses the first time derivative of the vector field along the trajectory. By iterating, we obtain Lemma 7 which expresses the k-th time derivative of the vector field along the trajectory. By upper bounding the r.h.s. in Lemma 7, we obtain Lemma 8 which bounds the k-th time derivative of the vector field along the trajectory. This bound means that the vector field along the trajectory has a small k-th derivative, therefore is well approximated by a polynomial. To derive our method, we can therefore approximate the vector field along the trajectory by a polynomial, and run collocation with this polynomial."
>
> **In addition, the lack of empirical validation remains a notable weakness. The rebuttal does not sufficiently justify this point, as it does not discuss the practical challenges of implementing the algorithm. This raises concerns for me that the method may perform poorly in practice, which in turn casts doubt on its theoretical value.**
>
> Let us try to clarify why we do not want to include simulations in this paper, and we hope that this will affect your rating of our paper. Collocation is, with Runge Kutta and Euler, one of the most famous integrator used in numerical analysis. There is no challenge in implementing it (it is 10 lines of codes or less) and it performs well on toy examples as well documented in the numerical analysis literature.
>
> Therefore, adding simulations of Collocation in a toy model would look too simple for any numerical analyst or diffusion model applied researcher. We experienced this in the past. Relevant simulation should be on a real image or video generation task. But this is out of the scope of our paper, and left for future work. Our paper is methodological and as you said, the paper is already complex.
>
> *We hope you can reconsider your score in light of these precisions.*
>
> Best,
>
> Authors

---

### Official Review · Reviewer_htQS · 2025-10-31

**Soundness:** 2
**Presentation:** 2
**Contribution:** 2
**Rating:** 4
**Confidence:** 3

**Summary:**

In this paper, the authors introduce a new parallel sampler based on the colocation method [1] for diffusion models on Euclidean state spaces. The main contribution of the paper is to show a convergence bound that the iteration complexity scales logarithmically with the inverse accuracy (compared to polynomial for non-parallel samplers) and does not depend on the ambient dimension for a class of target distributions. The interest of the authors for the polylogarithmic dependency with respect to the inverse accuracy stems from the similar picture which exists for sampling where Metropolis-based correction can achieve poly-logarithmic (wrt the inverse accuracy) iteration complexity.

[1] Lee et al. -- Algorithmic theory of ODEs and sampling from well-conditioned logconcave densities

**Strengths:**

* The paper is well-written. I think the discussion and clarification regarding the high-accuracy situation in classical statistical sampling is useful for the readers. I could follow the paper with ease.

* The results are new and as far as I know the use of the colocation method [1] in diffusion models is novel.

[1] Lee et al. -- Algorithmic theory of ODEs and sampling from well-conditioned logconcave densities

**Weaknesses:**

* Assumption 1 seems extremely strong. I think the authors should discuss more the impact of this assumption. Especially with regards to the works of [1,2] which have weaker assumptions on the target distribution (I do understand that the samplers and results are different in those papers but I think it is important to consider the potential limitations of Assumption 1). It seems that it would be well discussed in the context of the manifold hypothesis.

* I believe that the related work section is a bit misleading. In particular I would have expected more discussion around the work of [5] which is a parallel method also exhibiting iteration complexity with poly-logarithmic dependency in the inverse of the accuracy. It also heavily relies on a parallel sampler. I am aware of the comment of the authors "Finally, we note that the collocation method (see Section 2.3) has been studied in the context of diffusions, but primarily as a way to parallelize the steps of the sampler Anari et al. (2023); Gupta
et al. (2024); Chen et al. (2024a), but not using low-degree polynomial approximation." This comment in my opinion tames the claims made in the introduction. Further clarification and comparison is required.

* There exists further work improving the dependency of the iteration complexity from linear with respect to the inverse accuracy. In particular, in [6] for instance the authors show bounds of the order of $O(1/\varepsilon^{1/K})$ where $K$ is the order of the samplers (see also the references therein).

* I do understand that this is a theoretical paper but given that the authors introduce a new algorithm it would be great if they could illustrate the methodology in low dimensional settings at least.

[1] Conforti et al. (2023) -- KL Convergence Guarantees for Score diffusion models under minimal data assumptions

[2] Benton et al. (2023) -- Nearly d-Linear Convergence Bounds for Diffusion Models via Stochastic Localization

[3] De Bortoli et al. (2022) -- Convergence of denoising diffusion models under the manifold hypothesis

[4] Azangulov et al. (2024) -- Convergence of diffusion models under the manifold hypothesis in high-dimensions

[5] Gupta et al. (2024) -- Faster Diffusion Sampling with Randomized Midpoints: Sequential and Parallel

[6] Li et al. (2025) -- Faster Diffusion Models via Higher-Order Approximation

**Questions:**

* The authors only discuss the ODE case, what about the SDE one? The introduction of stochasticity is very useful in practice.

* Assumption 4 is very strong. Could you discuss this assumption and refer to other papers which might use a similar assumptions (if they exist?).

* I haven't put this concern in the Weaknesses section because it might simply be that I misunderstood part of the paper. Something that is unclear to me is the number of calls required to the model. It seems to me that the number of calls to the model in the Picard iteration is $DN$. Then in the whole diffusion sampler we use $n$ iterations resulting in $NDn$ (by the way there is a typo in Algorithm 2, line2, $k=q$ should be $k=0$). Is $D=d$? I am having trouble understanding the difference between these two quantities. If that is indeed the case then if we require $d$ calls to the model then the method is **highly** impractical in high dimension.

* I am a bit confused as to what is relevant to the "parallel" part of the sampler. In [1] for instance multiple steps are denoised during the Picard iterations. It doesn't seem to be the case here for me. Could you clarify?

* Could the method be applied to provide convergence bounds  for algorithms leveraging other parallel procedures such as speculative sampling [2]?

* Some lemmas do not compile in the supplementary material

[1] Shih et al. (2023) -- Parallel Sampling of Diffusion Models

[2] De Bortoli et al. (2025) -- Accelerated Diffusion Models via Speculative Sampling

---

> ### Author Response · Authors · 2025-11-14
> **Addressing the weaknesses**
>
> Thank you for your review. In this comment we address the potential weaknesses of our work.
>
> **Assumption 1 seems extremely strong. I think the authors should discuss more the impact of this assumption. Especially with regards to the works of [1,2] which have weaker assumptions on the target distribution (I do understand that the samplers and results are different in those papers but I think it is important to consider the potential limitations of Assumption 1). It seems that it would be well discussed in the context of the manifold hypothesis.**
>
> We believe our Assumption 1 is actually quite natural. For instance, if one wishes to sample from a general distribution $p$, even one supported on a low-dimensional manifold, it is common in both theory and practice [1,2] to consider early stopping the diffusion sampler, which amounts to sampling from $p$ convolved with a small amount of Gaussian noise. Of course, our result is formally incomparable with results that only assume, say, bounded second moment for the distribution, but our guarantee is exponentially faster than those works in its dependence on the target accuracy.
>
> **I believe that the related work section is a bit misleading. In particular I would have expected more discussion around the work of [5] which is a parallel method also exhibiting iteration complexity with poly-logarithmic dependency in the inverse of the accuracy. It also heavily relies on a parallel sampler. I am aware of the comment of the authors "Finally, we note that the collocation method (see Section 2.3) has been studied in the context of diffusions, but primarily as a way to parallelize the steps of the sampler Anari et al. (2023); Gupta et al. (2024); Chen et al. (2024a), but not using low-degree polynomial approximation." This comment in my opinion tames the claims made in the introduction. Further clarification and comparison is required.**
>
> We would like to clarify that the speedups achieved in our work are not coming from collocation specifically, but from a subtle interplay between *low-degree approximation* and collocation. Note that in the parallel sampling papers, while they get polylog(1/eps) round complexity, the total number of calls to the score oracle is still poly(1/eps). For instance, Algorithm 7 (line 1b) in [5] breaks up into O(eps)-length intervals in a given collocation window. Inside these intervals they make polylog oracle calls. Therefore the total work (i.e., number of oracle calls) in [5] is still polynomial (larger than 1/eps), whereas in our algorithm the total work is polylogarithmic. In fact, this is the whole point behind using low-degree polynomial approximation, which allows us to take larger steps in the collocation windows and avoid poly(1/eps) dependence.
>
>
> **There exists further work improving the dependency of the iteration complexity from linear with respect to the inverse accuracy.**
>
> Thank you. We will add the missing refs. To our knowledge they do not achieve logarithmic complexity.
>
> **I do understand that this is a theoretical paper but given that the authors introduce a new algorithm it would be great if they could illustrate the methodology in low dimensional settings at least.**
>
> We understand this concern but we prefer to keep our paper theoretical. We already convey several complex ideas which are new. Experimentation, engineering tricks etc. are left for future work.

---

> ### Author Response · Authors · 2025-11-16
> **Answer to questions**
>
> **The authors only discuss the ODE case, what about the SDE one? The introduction of stochasticity is very useful in practice.**
>
> Collocation is possible for SDE, however a complexity analysis will likely involve some dependency in the dimension d (because of the variance of the Gaussian noise), whereas our result does not depend explicitly on the dimension.
>
>  **Assumption 4 is very strong. Could you discuss this assumption and refer to other papers which might use a similar assumptions (if they exist?)**
>
> We proved in Lemma 36 that the true score is Lipschitz along the trajectory. Therefore, Assumption 4 is not really an assumption but a design choice: the user should use a Lipschitz neural net to learn the score. This makes sense because the true score is Lipschitz.
>
> References where Assumption 4 is used include:
>
> Chen, S., Chewi, S., Lee, H., Li, Y., Lu, J., & Salim, A. (2023). The probability flow ode is provably fast. Advances in Neural Information Processing Systems, 36, 68552-68575.
>
> Gupta, S., Cai, L., & Chen, S. (2024). Faster diffusion-based sampling with randomized midpoints: Sequential and parallel. arXiv e-prints, arXiv-2406.
>
>
> **I haven't put this concern in the Weaknesses section because it might simply be that I misunderstood part of the paper [...]**
>
> The total number of calls is indeed $NDn$ and the total number of iterations is $Nn$. The values of $N$ and $n$ are specified in Theorem 12, which counts the number of iterations in Algo 2. The value of $D$ is $log(1/\epsilon_{err})$, see line 1714. Thanks for the typo.
>
> **I am a bit confused as to what is relevant to the "parallel" part of the sampler. In [1] for instance multiple steps are denoised during the Picard iterations. It doesn't seem to be the case here for me. Could you clarify?**
>
> **Could the method be applied to provide convergence bounds for algorithms leveraging other parallel procedures such as speculative sampling [2]?**
>
> Could you please clarify the last two questions? Note that our algorithm is purely sequential, even though developing a parallel version would be an interesting future direction.
>
> **Some lemmas do not compile in the supplementary material**
>
> Thanks for spotting these errors. We submitted the Appendix without the main paper as supplementary material, this is where the errors come from.
>
>
> *We hope you will be able to reconsider your score in light of the clarifications we provided.*

---

### Official Review · Reviewer_Ppqb · 2025-11-03

**Soundness:** 3
**Presentation:** 3
**Contribution:** 2
**Rating:** 4
**Confidence:** 2

**Summary:**

This paper proposes a new solver of diffusion models based on the collocation method to solve ODE, and proves that its sampling complexity is logarithmically in $1/\epsilon$ for $\epsilon$ error. This complexity is improved over  that of the classical diffusion samplers. The paper provides theoretical construction of the proposed sampler and relevant proofs. The key assumption relies on some smoothness property of the score function, which allows a low-degree polynomial approximation. This approximation motivates the use of collocation ODE solver, which has exponential convergence in small time windows. Overall, the sampler is proven to have polylogarithmically complexity.

**Strengths:**

- The paper is well-written and easy-to-follow.

- This work seems to be the first analysis to give a polylog-complexity diffusion sampler.

**Weaknesses:**

The contribution of this work is primarily theoretical and the practical implication is not clear. The proposed sampler appears to be a mathematical construction rather than a practical algorithm, as it relies on a polynomial approximation of the score function that is not tractable in real diffusion inference settings. In this sense, the current title and abstract may be somewhat misleading, as they somewhat suggest the existence of an actual sampler rather than a theoretical construction.

**Questions:**

- What is the “aforementioned Gaussian mixture setting” mentioned in the last paragraph of page 2?

- Algorithm 2 line 2, "for k=q,...,n-1,n", what is q?

---

> ### Author Response · Authors · 2025-11-14
> **Important clarification**
>
> We thank you for your review.
>
> **The contribution of this work is primarily theoretical and the practical implication is not clear. The proposed sampler appears to be a mathematical construction rather than a practical algorithm, as it relies on a polynomial approximation of the score function that is not tractable in real diffusion inference settings. In this sense, the current title and abstract may be somewhat misleading, as they somewhat suggest the existence of an actual sampler rather than a theoretical construction.**
>
> We believe there is a misunderstanding here. We are not showing the "existence" of a sampler. We prove that the score is well-approximated by a low-degree polynomial as a function of time, but the way we leverage this polynomial approximation in the sampler is entirely algorithmic. As long as one has black-box access to estimates of the score functions, we can run our Algorithm 2. In particular, the basis of low-degree polynomials we work with is **independent** of the score estimate.
>
> In general, the polynomials are entirely explicit and specified by the interpolation condition on P: they should interpolate the $c_i \in [0,H]$. In particular, one can simply take them to be Lagrange polynomials built on Chebyshev nodes $c_i$ (translated to be between 0 and T) which is the golden standard for polynomial interpolation.
>
> *As this makes an essential difference to the practical implications of our paper, we hope you will be able to raise your score.*
>
> **What is the “aforementioned Gaussian mixture setting” mentioned in the last paragraph of page 2?**
>
> The “aforementioned Gaussian mixture setting” is mentioned above, at line 74.
>
>  **Algorithm 2 line 2, "for k=q,...,n-1,n", what is q?**
>
> This is a typo. It should be k=1,...,n-1,n.

---

### Author Response · Authors · 2025-12-03
**Overview of discussion**

## Summary
We would like to reiterate the significance and novelty of our work: we give the first **diffusion-based sampler which provably achieves a number of steps which is *logarithmic* in the target accuracy** (prior bounds were polynomial) and **does not explicitly depend on the dimension**. The mathematical machinery is a sharp departure from prior works, relying upon a subtle and entirely new interplay between the collocation method and, crucially, the low-degree approximability of the drift as a function of time, which we also believe to be of independent interest.

Two reviewers (Reviewer Ppqb and Reviewer htQS) who rated our submission with 4's did not reply to our rebuttal before the discussion freeze. Below is a summary of our answers to these two reviewers.

## Reviewer Ppqb:
Their assessment was based on the following criticism which turns out to be a **simple misunderstanding** of our result: “The proposed sampler appears to be a mathematical construction rather than a practical algorithm, as it relies on a polynomial approximation of the score function that is not tractable in real diffusion inference settings.”

Contrary to the reviewer’s claims, we are not showing mere "existence" of a sampler. **We prove the score is well-approximated by a low-degree polynomial in time, but the way we leverage this is entirely algorithmic.** As long as one has black-box access to score estimates, we can run our Algorithm 2. In particular, the basis of polynomials we work with is explicit: One can simply take them to be Lagrange polynomials built on Chebyshev nodes, suitably translated, a standard recipe for polynomial interpolation.

## Reviewer htQS
Their assessment was based on the following issues, one of which was a misunderstanding of the central premise of our proof and the rest of which we believe we adequately address:
- **Connection to parallel sampling:** The reviewer notes that previous works on parallel sampling with diffusion models achieved *parallel complexity* that depended logarithmically in 1/eps and also used the collocation method, and they surmised that our result follows along similar lines. We stress **this crucially overlooks the main technical point of our work. Our speedups are not coming from collocation alone, but from a subtle interplay between low-degree approximation and collocation.** Note that in the parallel sampling papers the reviewer mentions, the total number of calls to the score oracle is still poly(1/eps), whereas in our algorithm this is polylogarithmic. In fact, this is the whole point behind using **low-degree polynomial approximation, which allows us to take larger steps in the collocation windows and avoid poly(1/eps) dependence.**
- **Strength of assumptions:** The reviewer viewed our Assumptions 1 and 4 as strong, but we would like to push back on this, as we did in our rebuttal. First, Assumption 1, which is that the distribution is a compactly supported density plus a tiny amount of noise, is quite natural. If one wishes to sample, e.g., from a distribution supported on a compact manifold, **it is common in both theory and practice to consider early stopping the sampler, which is precisely meant to sample from a compactly supported density plus a small amount of noise**, which already represents good quality samples. While this is technically not as general as the most general assumptions in the literature, it is an exceedingly small cost to pay in exchange for an *exponential* speedup in the dependence on target accuracy. Assumption 4 is even more minor: it requires the score estimate be Lipschitz. Firstly, the true score is proven to be Lipschitz (Lemma 36), so Assumption 4 is more a design choice than an assumption (i.e., the user should use a Lipschitz neural network to learn the score because the score is Lipschitz). Secondly, the **assumption that the estimated score is Lipschitz is common in the literature**, see e.g., works of [Chen et al. ‘24, “The probability flow ODE is provably fast”], [Gupta et al. ‘25, “Faster diffusion-based sampling with randomized midpoints: Sequential and parallel”], and [Zhang et al. ‘25, “Sublinear iterations suffice even for DDPMs”].
- **Experiments**: The reviewer takes issue with the lack of experiments. We view this to be an unreasonable criticism: by now there have been a large number of papers on diffusion model theory that have appeared at NeurIPS/ICML/ICLR, almost none of which include numerical experiments (**in fact 5 of the 6 papers that the reviewer themself cited do not**). As a broader point, we view this kind of criticism to be unnecessarily exclusive of theory contributions to the ML community; in contrast, we certainly don't require experimental papers to provide rigorous end-to-end proofs to be accepted.

Finally, we note there are some minor points raised by reviewer htQS (e.g. one missing citation, and a small technical misunderstanding on their part) which we also address in the rebuttal.

---

### Meta-Review · Area_Chair_LHy5 · 2026-01-04

**Summary:**

The paper proposes a new diffusion solver based on the collocation method, offering a theoretical guarantee of logarithmic iteration complexity in $1/\epsilon$ with dimension-free bounds (scaling instead with the effective radius).While the Area Chair and reviewers acknowledge the theoretical novelty of achieving high-accuracy sampling in this regime, the consensus leans towards rejection due to a significant gap between the theoretical claims and practical validation. The primary concerns driving this decision are:

(1) **Lack of Empirical Validation.** Multiple reviewers (htQS, Ppqb, r9Fo, Z2pA) pointed out the complete absence of experiments. While the authors argued that the paper is purely methodological and that implementation is trivial ("10 lines of code"), the reviewers remained concerned that the method—which relies on polynomial approximations of the score—might face stability issues or intractability in real-world high-dimensional settings. The refusal to provide even a toy simulation to validate the "dimension-free" claims was a major sticking point.

(2) **Strength of Assumptions.** Reviewers expressed concern regarding the restrictive nature of Assumption 1 (bounded support plus noise) and the requirement for score estimation errors to have sub-exponential tails. There is skepticism about whether these assumptions hold sufficiently in practice to make the theoretical bounds relevant.

(3) **Readability and Accessibility.** Reviewer Z2pA noted that the technical lemmas were extremely difficult to parse, a sentiment that persisted even after the authors provided high-level summaries.Ultimately, while the mathematical construction is interesting, the submission feels incomplete without evidence that the proposed solver is robust outside of the idealized theoretical framework.

**Reviewer Concerns:**

**Concerns Addressed**

(1) Constructive vs. Existential. Reviewer Ppqb initially misunderstood the method as a mere proof of existence. The authors successfully clarified that the algorithm is constructive and relies on standard polynomial interpolation (e.g., Chebyshev nodes).

(2) Clarification of Norms. The authors addressed Reviewer Z2pA's confusion regarding the definitions of various norms (e.g., $||\cdot||_{p,\infty}$ vs Euclidean) used throughout the lemmas.

(3) Parallel Sampling Distinction. The authors provided a reasonable rebuttal to Reviewer htQS regarding the distinction between their method and prior parallel sampling works, clarifying that their efficiency gain comes from low-degree approximation rather than parallelization alone.

**Concerns Outstanding**

(1) Empirical viability (major). This remains the most significant outstanding issue. Reviewer Z2pA explicitly stated in the post-rebuttal discussion that the "lack of empirical validation remains a notable weakness" and that the rebuttal did not justify excluding it. The concern is that the polynomial approximation of the score might be unstable in practice, rendering the "logarithmic complexity" moot.

(2) Restrictive assumptions (minor). Reviewer htQS’s concern regarding Assumption 1 (compact support) and the strong assumption on the tail behavior of the score error (Reviewer r9Fo) remain points of friction. The authors argue these are necessary for the theory, but the reviewers remain unconvinced of their generality.

(3) Complexity/Readability (minor). Despite the authors' explanations, Reviewer Z2pA maintained that the proofs are "horrible" to look at and difficult for a diverse audience to follow, suggesting the paper needs significant restructuring to be accessible.

**Reviewer Scores:**

(1) Reviewer Z2pA (Score: 6->4/6): This reviewer actively participated in the discussion. The reviewer explicitly stated they would "maintain my score" of 6 (borderline) but noted they lean toward rejection if forced, citing the persistent lack of empirical validation and difficulty of the proofs. If the discussion had continued, their score likely would have remained a 6 or dropped to a 5, as they were unconvinced by the authors' refusal to add experiments.

(2) Reviewer r9Fo (Score: 6->6): This reviewer also engaged and explicitly stated, "I will maintain my rating" after the rebuttal. They acknowledged the theoretical novelty but listed the strong assumptions and lack of experiments as weaknesses. It is unlikely their score would have changed.

(3) Reviewer Ppqb (Score: 4->4): While the authors clarified the "existence" misunderstanding, Ppqb's fundamental critique was that the method appears to be a "mathematical construction rather than a practical algorithm." Given the authors' staunch refusal to include experiments, Ppqb would likely have maintained a score of 4, viewing the paper as theoretically interesting but practically unproven.

(4) Reviewer htQS (Score: 4->4): They had strong reservations about the related work citations and the restrictiveness of Assumption 4. The authors' rebuttal was somewhat dismissive of the request for experiments ("unreasonable criticism"). It is highly probable that htQS would have maintained a 4, as their concerns about the "extremely strong" assumptions were defended by the authors rather than mitigated.

---

### Decision · Program_Chairs · 2026-01-26

Reject